# An inductive bias for slowly changing features in human reinforcement learning

**Noa L. Hedrich** [1,2,3]*, **Eric Schulz**[4,5], **Sam Hall-McMaster**[1,6‡], **Nicolas W. Schuck** [1,2,7‡]*

**1** Max Planck Research Group NeuroCode, Max Planck Institute for Human Development, Berlin, Germany, **2** Institute of Psychology, Universität Hamburg, Hamburg, Germany, **3** Einstein Center for Neurosciences Berlin, Charité Universitätsmedizin Berlin, Berlin, Germany, **4** Max Planck Research Group Computational Principles of Intelligence, Max Planck Institute for Biological Cybernetics, Tübingen, Germany, **5** Helmholtz Institute for Human-Centered AI, Helmholtz Center Munich, Neuherberg, Germany, **6** Department of Psychology, Harvard University, Cambridge, Massachussets, United States of America, **7** Max Planck UCL Centre for Computational Psychiatry and Ageing Research, Berlin, Germany

‡These authors are joint senior authors on this work.
* noa.hedrich@uni-hamburg.de (NLH); nicolas.schuck@uni-hamburg.de (NWS)

**Data Availability Statement:** All code related to this work has been made openly available at https://github.com/noahedrich/slow_prior. The data collected in the experiment, as well as the model fitting data derived from the collected data,

## Abstract

Identifying goal-relevant features in novel environments is a central challenge for efficient behaviour. We asked whether humans address this challenge by relying on prior knowledge about common properties of reward-predicting features. One such property is the rate of change of features, given that behaviourally relevant processes tend to change on a slower timescale than noise. Hence, we asked whether humans are biased to learn more when task-relevant features are slow rather than fast. To test this idea, 295 human participants were asked to learn the rewards of two-dimensional bandits when either a slowly or quickly changing feature of the bandit predicted reward. Across two experiments and one preregistered replication, participants accrued more reward when a bandit's relevant feature changed slowly, and its irrelevant feature quickly, as compared to the opposite. We did not find a difference in the ability to generalise to unseen feature values between conditions. Testing how feature speed could affect learning with a set of four function approximation Kalman filter models revealed that participants had a higher learning rate for the slow feature, and adjusted their learning to both the relevance and the speed of feature changes. The larger the improvement in participants' performance for slow compared to fast bandits, the more strongly they adjusted their learning rates. These results provide evidence that human reinforcement learning favours slower features, suggesting a bias in how humans approach reward learning.

## Author summary

Learning experiments in the laboratory are often assumed to exist in a vacuum, where participants solve a given task independently of how they learn in more natural circumstances. But humans and other animals are in fact well known to "meta learn", i.e. to leverage generalisable assumptions about *how to learn* from other experiences. Taking inspiration from a well-known machine learning technique known as slow feature

are openly available at OSF https://osf.io/b9n5y/, with the DOI: 10.17605/OSF.IO/B9N5Y.

**Funding:** This research was supported by an Einstein Centre for Neurosciences PhD Fellowship awarded to NLH, an Alexander von Humboldt Fellowship awarded to SHM, a European Research Council Starting grant (ERCStG-REPLAY-852669), funding from the Max Planck Society (M.TN.A. BILD0004), and funding from the BMBF (Excellence Strategy of the Federal Government and the Länder) awarded to NWS. The funders had no role in study design, data collection and analysis, decision to publish, or preparation of the manuscript.

**Competing interests:** The authors have declared that no competing interests exist.

analysis, we investigated one specific instance of such an assumption in learning: the possibility that humans tend to focus on slowly rather than quickly changing features when learning about rewards. To test this, we developed a task where participants had to learn the value of stimuli composed of two features. Participants indeed learned better from a slowly rather than quickly changing feature that predicted reward. Computational modelling of participant behaviour indicated that participants had a higher learning rate for slowly changing features from the outset. Hence, our results support the idea that human reinforcement learning reflects a priori assumptions about the reward structure in natural environments.

## Introduction

A remarkable amount of information is reaching our senses at any given time, yet often only a small subset of it is relevant to our current goal. Determining which aspects of our environment are relevant is therefore a crucial challenge for learning goal-directed behaviour. But addressing this challenge is hard. The space of possibilities is often too large to be explored fully within the time limits we need to consider, and yet limiting attention to only a subset of features risks ignoring relevant information [1, 2]. One approach to this problem is to not learn every problem anew, but instead use knowledge of properties that have been relevant in the past as a starting point, in the form of so-called priors, also known as inductive biases [3–7]. Here, we study the role of one such prior in human learning, namely a bias to focus learning on slowly changing features in our environments, and their potential association to rewards.

Analogous to the concept of a 'prior' in Bayesian statistics, priors are pre-existing beliefs about the underlying structure of an environment, based on generalised past experiences or evolutionary transmission [3, 8]. Previous research has shown that priors can expedite the learning process by focusing information processing on what is common across many environments [4, 9, 10]. The resulting decision-making biases are numerous [10–13] and can for instance be observed in the form of adaptive heuristics that reflect constraints on time or resources [14], or in the form of visual illusions that reflect the simplifying assumptions of our visual system, such as that light tends to come from above [15]. Studying useful priors for representation learning is also an active field of development in artificial intelligence [8, 16–18], in particular for reinforcement learning (RL), where knowledge about which actions maximise reward and minimise punishment is acquired through a trial-and-error process [19]. While the RL framework has been very successful in furthering our understanding of learning and decision-making, [20–23], it becomes notoriously inefficient in high dimensional environments [19]. This problem can be alleviated through a process known as representation learning, where learning is limited to a subset of features that help predict future rewards, known as task states [19, 24–28]. The difficulties of learning the state space for each new problem *de novo* have been widely recognized [29], underscoring the potential benefit of leveraging prior knowledge.

A useful prior for reinforcement learning should therefore help quickly build appropriate task states from rich perceptual observations in novel environments [8, 30]. A characteristic shared across many environments is that the causal process generating observations develops on a slower timescale than the sensory signals we observe [31–33]. For example, the appearance of a ball flying toward you in a park might fluctuate rapidly as it passes through patches of sun and shade, but its true colour will remain unchanged. Similarly, other relevant properties

such as its speed and trajectory will change in a slower, continuous manner compared to low-level perceptual features. In short, signal tends to vary more slowly than noise [34]. Consequently, slow features are both more predictable, making them easier to learn from, and useful for predicting future observations, therefore warranting greater attention [35, 36]. It follows that a way to extract features relevant to building task states, while remaining impartial to the exact nature of those features or the causal process underlying the perceptual observations, is to focus on slowly changing features. Indeed, some research has shown that humans have a bias toward perceiving slower speeds in the spatial domain [34, 37, 38]. This idea has gained more traction in machine learning, where a slowness prior has been shown to enable the discovery of task states from raw observations [8, 28, 39, 40].

A well-known implementation of this prior is Slow Feature Analysis (SFA), an unsupervised learning algorithm that reduces the dimensionality of its input by identifying slowly changing dimensions in the data [31, 41, 42]. SFA first isolates independent components in the data and then extracts those components that change slowly, under the premise that slower features are more meaningful representations of the data [31]. This insight has been shown to be relevant for RL, for instance in the context of a spatial learning task where SFA can provide a effective representation learning mechanism [43]. The same paper showed that the SFA agent produced similar learning trajectories to rats solving a comparable task, underscoring the relevance of a slowness prior for animals. Theoretical research also demonstrated that extracting slow features can explain the activity of complex cells in the visual cortex, the formation of allocentric spatial maps in the hippocampus and can be implemented in a biologically plausible network [44–48]. Hence, a slowness prior promises a domain-general and biologically plausible way to extract task states from environmental input.

Despite its potential for representation learning and the abundance of research in the machine learning domain, studies on the slowness prior in human reinforcement learning are largely lacking. Here we explored the idea that humans rely on a slowness prior during reinforcement learning. We developed a novel decision-making task, in which participants had to repeatedly learn which of two stimulus features predicted reward. We manipulated the speed of change of the features and asked whether participants were faster to learn when the relevant feature changed slowly versus when it changed quickly. Across two experiments and one pre-registered replication, as well as extensive model comparison, our results indicate that they do. This finding enriches our understanding of human inductive biases in RL and can prompt further studies about other such biases in human learning, as well as inform artificial intelligence about how to best build human-like agents.

## Results

We investigated whether humans have a prior to preferentially process slowly changing features of the environment that impacts reinforcement learning. We hypothesised that given such a prior, participants should be better at learning the task if reward-predictive features changed slowly, rather than quickly. To test this, we developed a task that required participants to learn the rewards associated with a set of visual stimuli characterized by two features, a colour and a shape (Fig 1A). During each trial of learning, participants saw a stimulus composed of both features and decided between rejecting or accepting the stimulus. While rejecting always led to a fixed reward of 50 coins, accepting led to reward between 0 to 100 coins that was higher than 50 for half of all stimuli. Across trials, the two features changed independently and with different speeds: one feature changed slowly (e.g., participants saw relatively similar shapes from trial to trial), while the other feature changed quickly (e.g., participants saw relatively distinct colours from trial to trial, Fig 1A). Our core manipulation was that in each block

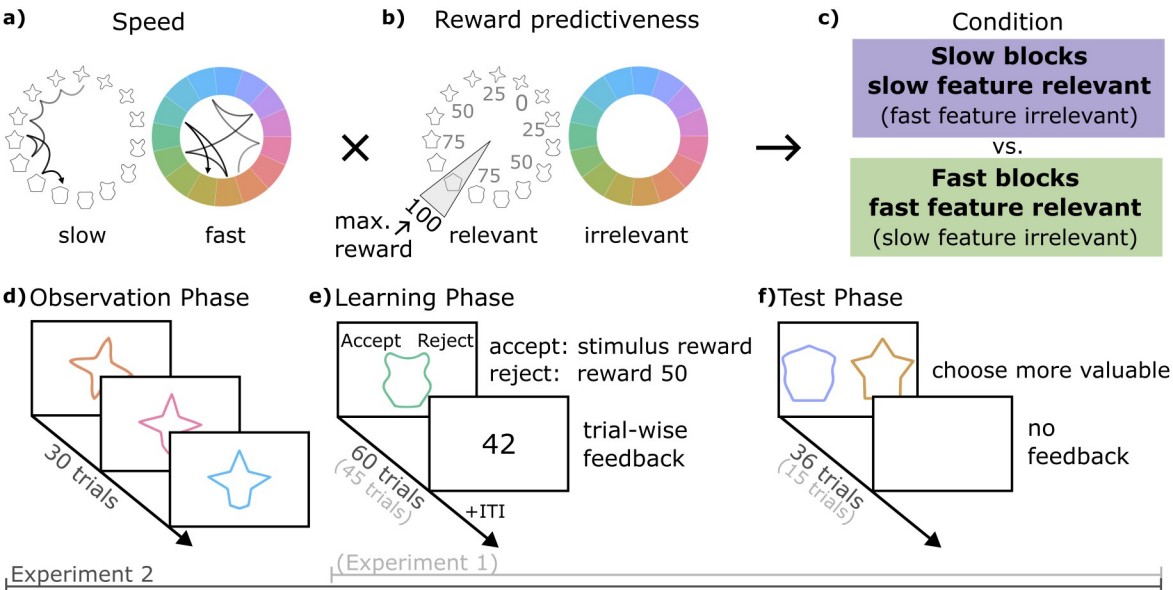

**Fig 1. Continuous reward features learning task. A:** The two stimulus features and their possible speeds. Each jump of the arrows indicates the change in the feature from one trial to the next. The slow feature (here: shape) changes gradually, while the fast feature (here: colour) changes randomly. The feature-speed mapping is only for illustration, in each block, either shape or colour could change slowly. **B:** The mapping of reward onto the relevant feature space. The relevant feature (here: shape) determines the stimulus reward. The closer the stimulus shape is to the maximum reward location, the higher the reward. The irrelevant feature (here: colour) was uncorrelated with reward. The feature-reward mapping is only for illustration, in each block, either shape or colour could be relevant and the maximum reward location changed. **C:** How feature speed and reward predictiveness were combined to form slow and fast blocks. Note that which feature was slow/relevant was counterbalanced across blocks. **D-F:** Schematic of the three phases in each task block. In experiment 1, the observation phase **D** was omitted and the learning and test phases were shorter.

either the slowly-changing or the fast-changing feature was reward-predictive, while the other had no relation to reward (relevant and irrelevant feature, respectively). The relevant feature had a fixed relation to reward in each block, with the maximum reward of 100 assigned to one position and decreasing rewards assigned to other positions based on their distance to the maximum. This split the circular feature space into two semicircles: high- and low-reward (Fig 1B). Hence participants had to learn which feature was reward-predictive in general, and which specific feature positions should be accepted vs rejected.

We first tested our hypothesis in two experiments, each with 50 participants (experiments 1 and 2), and then performed a preregistered replication of experiment 2 in a third sample of 195 participants (138 participants after exclusions, hereafter called the replication sample). The key difference between experiments 1 and 2 was that experiment 2 included a demonstration of stimulus changes before each block. The experiments also differed in the number of trials per block. Hence, in experiment 1 participants directly started reward learning, and could observe which feature changed fast vs. slow while they also had to observe the reward outcomes. In experiment 2 and the replication, we ensured participants knew how fast each feature would change *before* each block by displaying a sequence of 30 trials without reward that participants observed passively before learning (*Observation phase*, see Fig 1D). Participants were not informed about which feature was relevant in either experiment but had to learn it in each block through trial and error from the *Learning phase*, as described above (experiment 1: 45 trials, experiment 2 and replication: 60 trials, Fig 1E). Due to the continuous reward structure, it was beneficial to generalise observed outcomes to nearby feature positions. We probed

generalisation of learned values at the end of each block in a *Test phase* in which participants were asked to choose the more valuable stimulus among pairs of stimuli not seen during learning, without feedback (experiment 1: 15 trials, experiment 2 and replication: 36 trials, Fig 1F).

Participants performed eight blocks in total. In half of the blocks the slow feature was reward-predictive (slow blocks), in the other half the fast feature was reward-predictive (fast blocks, Fig 1C). Within each of these conditions, colour and shape were assigned as the relevant feature an equal number of times.

The three experiments are accompanied by two preregistrations available on the Open Science Framework (OSF)(https://osf.io/6dy8f). The first preregistration was submitted prior to acquisition of any data. We used experiment 1 and 2 to refine the task and analysis. As a result, we deviated from the first preregistration, however note that the hypothesis, core tenets of the task and main analyses are maintained. To validate our results, we updated the preregistration to reflect the changes to the task and analysis and then conducted a direct replication of experiment 2 in strict adherence to this registration. This reregistration focused on 3 main effects: the effect of feature speed on (1) collected rewards and (2) accuracy in the learning phase and on (3) accuracy in the test phase. Three additional analyses that were less central were also included, namely logistic mixed effect models testing the impact of (1) reward and (2) relevant and irrelevant features on learning phase choices, as well as (3) the difference in reward between stimuli on test phase choices. Finally, we included the intent to conduct a meta-analysis across experiments in the registration. In the reregistration we defined a new performance-based exclusion criterion, which led to an effective sample size of 138 participants after exclusions in the replication. The computational models were not included in the preregistrations. For more details on the preregistration see Text A in S1 Materials.

## Participants learned feature rewards and generalised their knowledge

We first analysed participant choices to confirm learning of the feature-reward mapping. Participants' choice accuracy in the learning phase increased from around chance in the first ten trials of a block to significantly above chance in the last ten trials in all three experiment samples (experiment 1: start = 52%, end = 67%, $t(49) = 8.27$ $p < .001$, experiment 2: start = 51%, end = 74%, $t(49) = 13.26$ $p < .001$, replication: start = 51%, end = 75%, $t(137) = 26.91$ $p < .001$, Fig 2A). This increase in accuracy was accompanied by a gradual decrease in 'accept' choices throughout the learning phase, reducing from above 85% in the first ten trials to below 65% in the last ten trials across experiments (experiment 1: start = 85%, end = 64%, $t(49) = -10.28$ $p < .001$; experiment 2: start = 87%, end = 62%, $t(49) = -13.46$ $p < .001$; replication: start = 88%, end = 58%, $t(137) = -28.39$ $p < .001$, Fig 2B). Note that 'accept' choices allowed participants to gather information on stimulus values and therefore were necessary for exploration early in a block. Accordingly, participants learned with time to selectively reject low-value stimuli, while they continued to accept high-value stimuli (Fig 2C). We confirmed participants did not engage in simplified strategies by fitting two control models, one which captures possible biases for accept choices (Random Choice model), and one which can capture a bias for one of the response keys (Random Key model). These models did not explain participant choices well, compared to the learning models discussed below (see the computational models section and Methods for details). These results show that participants learned the feature-reward mapping.

We also found that participants could correctly identify the higher value stimulus in the test phase, in which previously unseen feature positions were presented, for which participants never witnessed reward feedback (mean accuracy significantly higher than the chance level of 50%, experiment 1: 71%, $t(49) = 12.26$ $p < .001$; experiment 2: 75%, $t(49) = 17.38$ $p < .001$; replication: 75% $t(137) = 32.04$, $p < .001$). Further, participant choice probabilities reflected

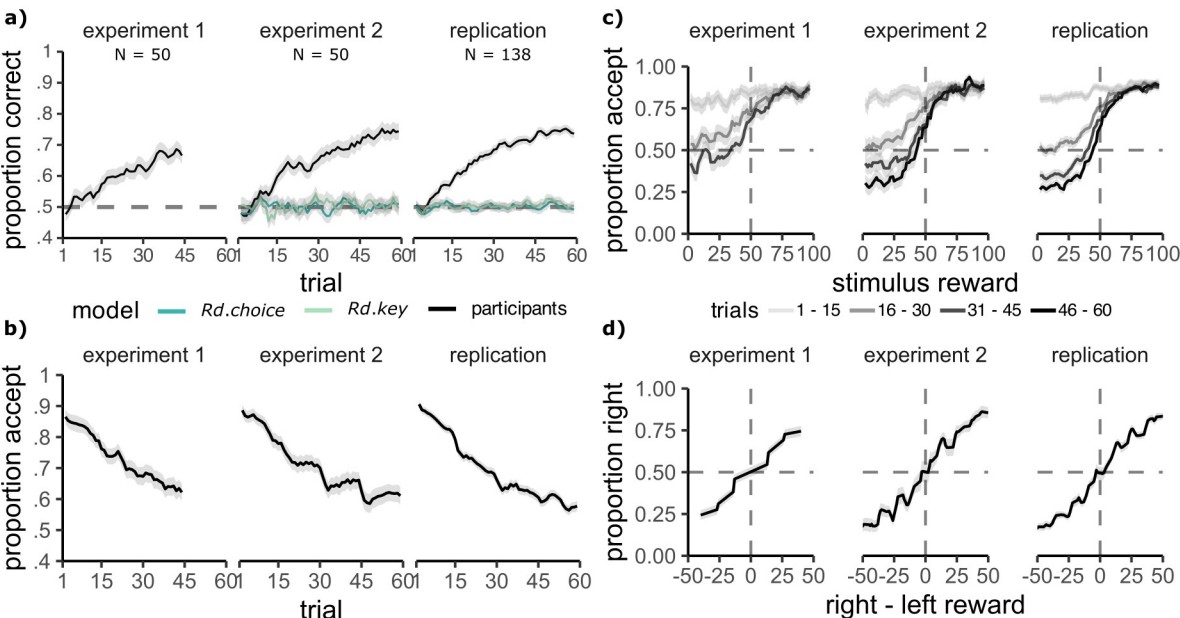

**Fig 2. Participants learned and generalised the feature reward mapping. A:** Proportion correct choices across trials increases in the learning phase. The behaviour of two control models which capture aspects of random behaviour are shown in blue/green colours. **B:** The proportion of accept choices in the learning phase reduces across trials. **C:** The proportion of accept choices depending on the true stimulus reward, for every 15 trials from the start to the end of the block. Participants learn to selectively reject low-value stimuli. **A-C:** Curves were averaged across 3 adjacent values. **D:** Proportion of choosing the right stimulus in the test trials, depending on the difference in value between the right and left stimulus, shows sensitivity to the true reward value. Curves were averaged across 5 adjacent values. Grey ribbons show the standard error of the mean.

true stimulus values (Fig 2D). Hence, our data suggests that participants generalised values successfully across task and stimulus differences between the two phases.

## Learning improved when the relevant feature changed slowly

Having established that participants learned and generalised well in our task, we turned to our main question, namely, whether reward learning and generalisation differed for slowly versus fast-changing features. All mixed effect models used the maximal random effects structure that converged. We first included all main effects and interactions between predictors in the fixed effects and then tested whether removing them impacted model fit. Predictors were z-scored and no response trials were excluded, see Methods for details. Full model descriptions including effect sizes, confidence intervals and best fit models in the case when the effect of interest did not contribute to the fit can be found in the Methods and in Tables A-N in S1 Materials.

**Improved learning.** We measured performance in the learning phase by subtracting the cumulative reward expected by chance (50 per trial) from the cumulative reward obtained by participants. In experiment 2 and the replication sample, in line with our hypothesis, the cumulative reward gain was higher in slow compared to fast blocks (preregistered main test 1, experiment 2: $M_S = 248.62 \pm 21.54$, $M_F = 217.57 \pm 22.43$, $t(49) = 2.17$, $p_{1-sided} = .017$, $d = 0.31$, N slow>fast = 33/50; replication: $M_S = 270.19 \pm 10.73$, $M_F = 242.95 \pm 10.96$, $t(137) = 2.65$, $p_{1-sided} = .004$, $d = 0.23$, N slow>fast = 86/138; Fig 3A and 3C). In experiment 1, in which the observation phase was omitted and blocks were shorter, participants accumulated higher reward on slow blocks, but the difference was not significant ($M_S = 128.11 \pm 14.03$, $M_F = 108.88 \pm 14.97$, $t(49) = 1.57$, $p_{1-sided} = .061$, $d = 0.22$, N slow>fast = 30/50, Fig 3A and 3C). The

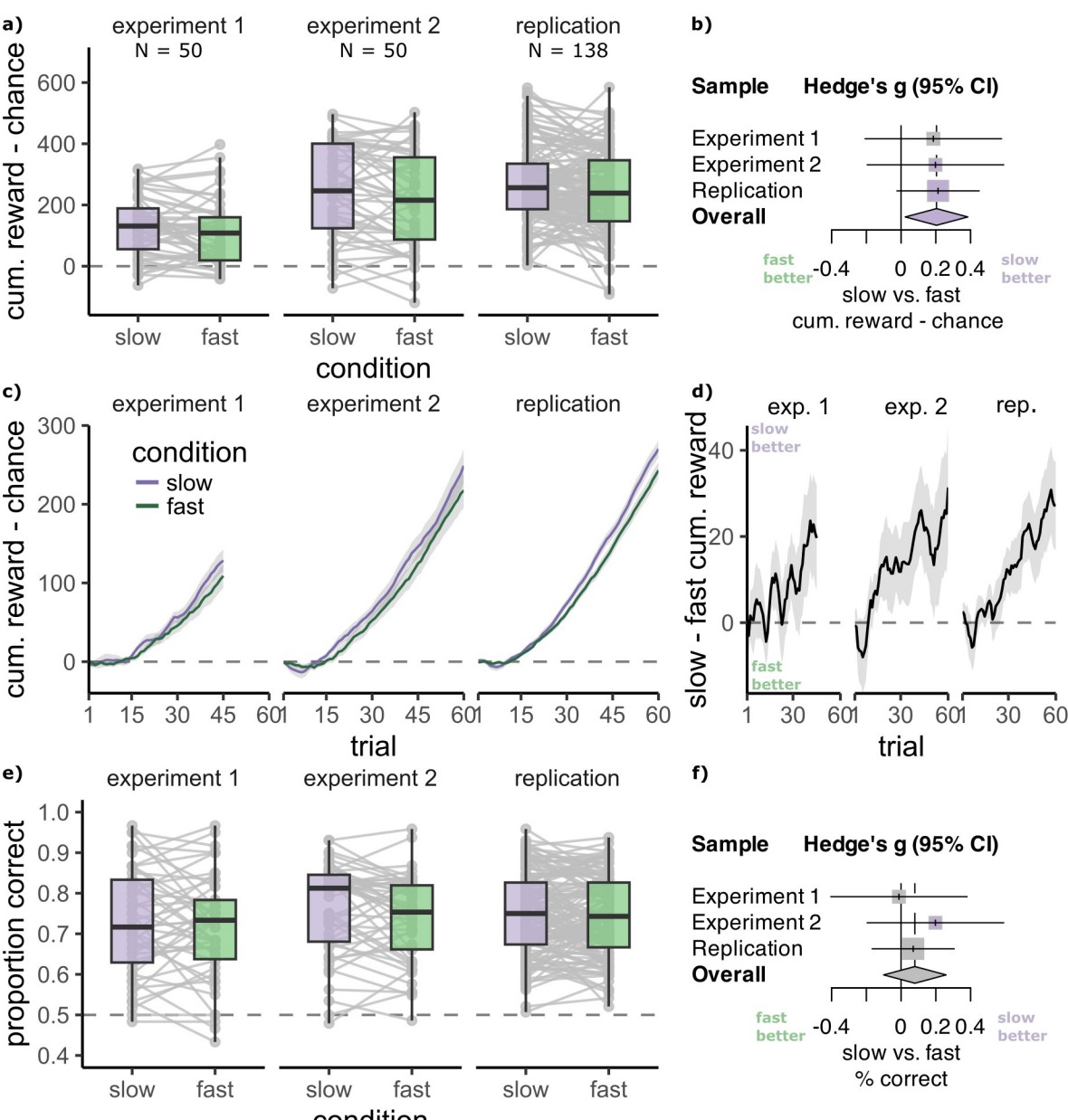

**Fig 3. Participants learned better in slow blocks. A:** Cumulative reward obtained in a block of the learning phase above a chance baseline of 50 per trial is higher in slow than in fast blocks in all three samples. Separately for blocks where the slow feature (purple) and fast feature (green) were relevant. Individual participant means in grey. **B:** Higher cumulative reward in slow compared to fast blocks is confirmed by a meta-analysis across experiments. **C:** Cumulative reward obtained relative to a chance baseline of 50 on each trial increases more rapidly in slow blocks in all three samples. Grey ribbons show the standard error of the mean. **D:** Visualisation of the difference in cumulative reward between slow and fast blocks across trials. **E:** Mean accuracy in the test phase is higher in slow than in fast blocks in experiment 2, but not experiment 1 and the replication. **F:** Meta-analysis results show that there is no consistent benefit in test phase generalisation for slow blocks.

learning benefit was also evident in an analysis of the average percent of correct choices in slow vs fast blocks in experiment 2 and the replication sample, but not in experiment 1 (preregistered main test 2, experiment 1: $M_S = 60\% \pm 1$, $M_F = 59\% \pm 1$, $t(49) = 1.14$ $p_{1-sided} = .130$, $d = 0.16$, N slow>fast = 28/50; experiment 2: $M_S = 65\% \pm 1$, $M_F = 64\% \pm 1$, $t(49) = 1.98$

$p_{1-sided}$ = .028, $d$ = 0.28, N slow>fast = 30/50; replication: $M_S$ = 66% ± 1, $M_F$ = 65% ± 1, $t$(137) = 1.99, $p_{1-sided}$ = .024, $d$ = 0.17, N slow>fast = 84/138, for the learning curves in slow and fast blocks see Fig A in S1 Materials). A logistic mixed effects model predicting participant choices with fixed effects for condition (slow/fast), trial number, stimulus value, and all two-way interactions showed that participant choices more closely reflected the stimulus values in the slow compared to the fast blocks in all samples (preregistered supplementary test 1, condition×value, experiment 1: $\beta$ = 0.094, 95% CI = [0.016 to 0.173], likelihood ratio test comparing to model without interaction: $X^2$(1) = 5.46, $p$ = .019; experiment 2: $\beta$ = 0.120, 95% CI [0.048 to 0.191], $X^2$(1) = 10.72, $p$ = .002; replication: $\beta$ = 0.106, 95% CI = [0.063 to 0.149], $X^2$(1) = 23.32, $p$ < .001). Additionally, meta-analytic results across the three experiment samples showed that reward was robustly higher on slow blocks of the learning phase (Hedge's g = 0.204, 95% CI = [0.024 to 0.385], Z = 2.22, p = .026, Fig 3B, for a forest plot with all statistics see Fig A in S1 Materials) with low heterogeneity between samples ($\tau^2$ = 0, $I^2$ = 0.0%, 95% CI = [0.0%; 89.6%], Q = 0.01, p = .993). In sum, participants were better at using stimulus values in slow relative to fast blocks and hence made more correct choices and accumulated more rewards. This lends support to the idea that participants benefited when the relevant feature was changing slowly.

Given that the slowness prior proposes that slow-changing features will be more likely to be considered relevant, we hypothesised that the lower reward and accuracy on fast blocks could result from incorrectly basing choices on the slow feature, even when it was irrelevant. To test this, we used the feature positions for both the relevant and irrelevant feature, trial number, and their interactions to predict participant choices separately for slow and fast blocks, using a logistic mixed effects model. We found no significant effect of the irrelevant feature on choices in either condition, across all samples (preregistered supplementary test 2, likelihood ratio test comparing to model without irrelevant feature terms, experiment 1: slow: $X^2$(4) = 5.07, $p$ > .05, fast: $X^2$(4) = 6.16, $p$ = .749; experiment 2: slow: $X^2$(4) = 3.34, $p$ > .05, fast: $X^2$(4) = 10.92, $p$ = .110; replication: slow: $X^2$(4) = 7.24, $p$ = .495, fast: $X^2$(4) = 2.73, $p$ > .05). Note that the preregistration states that we do not run model comparison for this analysis. While we do not exclude predictors based on model comparison, we do use it to assess the significance of predictors.

**No effect on generalisation.** We next asked whether a difference between slow and fast blocks was also evident in the test phase. While we did find that participants' accuracy was greater in slow versus fast blocks in experiment 2, we did not observe this effect in experiment 1 nor in the replication sample (preregistered main test 3, experiment 1: $M_S$ = 71% ± 2, $M_F$ = 72%±2, $t$(49) = 1.14, $p_{1-sided}$ = .130, $d$ = 0.16, N slow>fast = 22/50; experiment 2: $M_S$ = 76% ± 2, $M_F$ = 74% ± 2, $t$(49) = 1.85, $p_{1-sided}$ = .035, $d$ = 0.26, N slow>fast = 29/50; replication: $M_S$ = 75% ± 1, $M_F$ = 74% ± 1, $t$(137) = 0.96, $p_{1-sided}$ = .170, $d$ = 0.08, N slow>fast = 65/138; Fig 3E). The same picture emerged when modelling participant left/right choices in a logistic mixed effects model, with the condition, value difference and the condition×value difference interaction as predictors. We found that the true difference in value between the shown stimuli had a greater influence on choice in slow than in fast blocks in experiment 2, but not in the other two samples (preregistered supplementary test 3, condition×value difference: experiment 1: $\beta$ = −0.015, 95% CI = [-0.138 to 0.108], $X^2$(2) = 0.06, $p$ > .05; experiment 2: $\beta$ = 0.125, 95% CI = [0.038 to 0.212], $X^2$(2) = 7.93, $p$ = .010; replication: $\beta$ = 0.094, 95% CI = [-0.020 to 0.208], $X^2$(1) = 2.58, $p$ = .216). A meta-analysis did not show a consistent effect of feature slowness on accuracy in the test phase (Hedge's g = 0.080, 95% CI = [-0.100; 0.259], Z = 0.87, p = .386, Fig 3F, see also Fig A in S1 Materials, heterogeneity: $\tau^2$ = 0, $I^2$ = 0.0% 95% CI = [0.0%; 89.6%], Q = 0.56, p = .754). Hence, participants were not better able to infer and generalise the feature values in the test phase when the relevant feature had changed slowly during the learning

phase. In sum, we confirm the main analyses 1 and 2 and supplementary analysis 1 from our preregistration, but do not find evidence for main analysis 3 and supplementary analysis 2 and 3.

**Control analyses.** One possible concern regarding the interpretation of these effects is that the auto-correlation of reward outcomes could facilitate learning for slow but not for fast blocks. Our results speak against this interpretation. First, we tested a control model that ignored the stimulus features and simply learned a value estimate from successive reward outcomes (henceforth: Bandit Model). This model performed badly on the task and could not predict participant choices well (see model results below and Methods), suggesting that auto-correlation alone could not explain the differences in performance between slow and fast blocks. Second, we tested a control model that used a win-stay-lose-shift strategy (henceforth: WSLS Model) [49, 50]. This strategy can be helpful in slow blocks, where consecutive trials are likely to require the same choice, but not in fast blocks, where the correct choice is likely to change often. Indeed, this model performed well in slow blocks and badly in fast blocks (see Fig D in S1 Materials), but could not explain participant choices well (see model results below, and Methods).

## Computational models

To examine which mechanisms might underlie the difference in learning between the conditions, we fitted four reinforcement learning (RL) models to participant choices during the learning phase in experiment 2 and the replication study. Based on our behavioural findings above, all considered models sought to (a) reflect participants' learning from outcomes, (b) account for learning about which stimulus feature is relevant and which is not, (c) incorporate generalisation between stimuli of similar appearance, and (d) reflect participant's tendency to explore by accepting many stimuli when uncertainty is high. Our major aim was to test whether the learning process differed depending on whether participants learned about slow or fast-changing features, i.e. in slow vs fast blocks. To this end, we formulated a set of four models that embodied alternative hypotheses about how feature speed could affect learning, as described below (see Fig 4).

All models used linear function approximation and a Kalman filter to account for participants' generalisation and exploration behaviour, respectively (see Fig 4 and Methods). Briefly, each stimulus was converted into a 30-dimensional feature vector $\mathbf{x}_t$ that indicated which colour and shape stimulus on trial $t$ had (one entry for each of the 15 possible shapes and 15 colours). To reflect feature similarity across the circular stimulus space, a von-Mises distribution was centered around the true stimulus features, such that activation of node $i$ was determined by its distance from the node assigned to the true feature $t$:

$$x_{t,i} = \frac{e^{cos(d_{t,i})\kappa}}{\sum_{i=1}^{360} e^{cos(d_{t,i})\kappa}} \tag{1}$$

where $d_{t,i}$ is the distance between node $i$ and $t$ in radians and $\kappa$ determines the width (a.k.a. concentration) of the von-Mises distribution. We then modelled the expected value $V_t$ of a stimulus as the inner product of the feature vector $\mathbf{x}_t$ and the weight vector $\mathbf{w}_t$:

$$V_t = \mathbf{x}_t^T \mathbf{w}_t \tag{2}$$

and updated $\mathbf{w}_t$ after each accept choice to reflect the outcome $R_t$ of trial $t$ with a learning rate

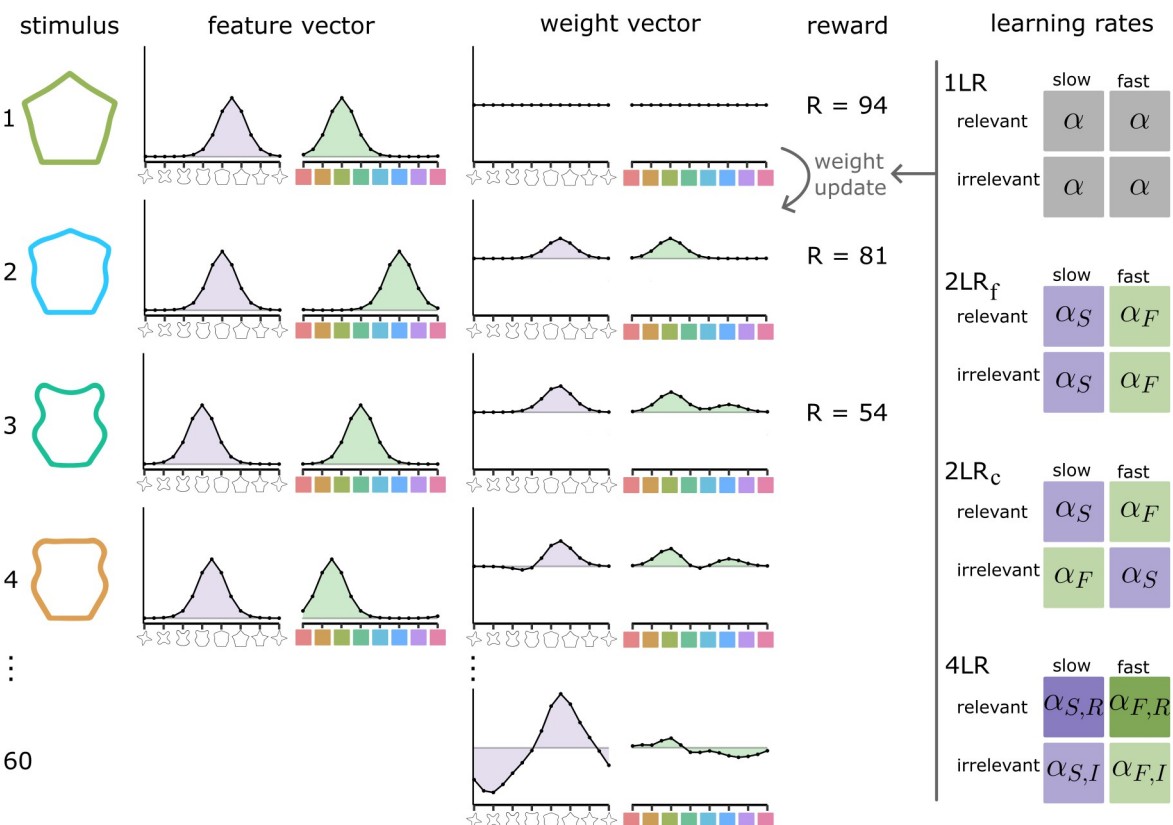

**Fig 4. Schematic of the RL models.** From left to right: A stimulus is converted to a feature vector, which is a distribution across neighbouring feature values. The feature vector is combined with the weight vector, which stores the value estimates. The resulting value for the stimulus is compared against the reward outcome. This reward prediction error is used to update the weight vector on each trial (shown as rows in the figure). By the end of the block (bottom row), the model learns a mapping between the relevant feature (in this case shape) and reward. The right column shows how the learning rates map onto the stimulus features. Learning models: one learning rate model (1LR), separate learning rates per slow/fast feature (2LR$_f$), separate learning rates per slow/fast condition (2LR$_c$) and the four learning rates model (4LR).

$\alpha$, as follows:

$$\mathbf{w}_{t+1} = \mathbf{w}_t + \alpha_t \mathbf{x}_t \left( R_t - V_t \right) \tag{3}$$

To account for exploration behaviour, we modelled participants' uncertainty, $U_t$, about the value of a stimulus on trial $t$ using a Kalman Filter. Akin to an upper confidence bound mechanism [51], the uncertainty was added to stimulus value in model choices, serving as an exploration bonus (see Methods for details):

$$V_{a,t} = V_t + c\, U_t \tag{4}$$

where $c$ mediates how strongly the exploration bonus is weighted at choice. The uncertainty $U_t$ also determined the learning rate on the current trial, $\alpha_t$. As the environment was stationary the uncertainty and learning rate reduced across trials. Finally, the model's choice was guided by the probability of the value of accepting, $V_{a,t}$, being larger than a normal random variable

centred on 50 (the value of rejecting), with standard deviation $\sigma$:

$$
\begin{aligned}
p(\text{accept}) &= P[X \leq V_{a,t}] \\
X &\sim N(50, \sigma^2)
\end{aligned}
\tag{5}
$$

While all of the four models reported here used the above-described mechanisms, they differed in whether they could adapt their learning rates to the slowness of the features, the relevance of the features to predict reward, or both (see Fig 4 right column). A baseline model used the same learning rate $\alpha$ for all conditions and features (one learning rate model, short 1LR). Hence, this model was indifferent to slowness and could not account for a difference in performance between the slow and fast blocks. A second model used separate learning rates for the slow vs. fast-changing feature ($\alpha_S/\alpha_F$), irrespective of whether the feature was relevant in a given block (feature learning rates model, 2LR$_f$). This model could account for the difference in performance between slow and fast blocks, but since it disregarded the relevance of the features for predicting reward it is an unlikely candidate to explain participant behaviour *a priori*. In a third model (condition learning rates model, 2LR$_c$), separate learning rates were used depending on whether the relevant feature was changing slowly ($\alpha_S$) or quickly ($\alpha_F$), but used the same learning rate for both features within the same block, regardless of their relevance. Finally, the fourth model had four separate learning rates for the slow and fast-changing features, when they were relevant and irrelevant (4LR model, learning rates $\alpha_{S,R}$, $\alpha_{F,R}$ vs $\alpha_{S,I}$, $\alpha_{F,I}$, respectively). This model could accommodate both differences in learning due to the slowness of the features and the reward structure of the task, for which reason we expected this model to predict participant choices best. The core aspect of these models is that they can assign different learning rates to the slow and the fast features or blocks *a priori*. Should the learning rate for the slow feature or condition be higher, this would lead to faster learning when the slow feature is relevant, and slower learning when the fast feature is relevant, effectively implementing our hypothesis.

We also considered models where the exploration parameter ($c$) or the decision noise ($\sigma^2$) could differ between conditions, however these models gave a worse account of participant behaviour (for details see Methods and Fig D in S1 Materials)

**All models can learn the task.**  To ensure that all models represent useful accounts of behaviour, we first fitted model parameters to maximise reward obtained by the model. This showed that given optimal parameters all learning models achieved a near-ceiling cumulative reward gain of around 600 coins per block, significantly above the cumulative reward expected by chance (all $p < .001$, theoretical maximum of clairvoyant agent: ca. 735 coins). In contrast, above mentioned Random Choice, Random Key, Bandit, and WSLS control models, were all significantly worse at the task (all $p < .001$, Fig 5A). In the test phase, the differences were even starker—only the learning models learned a mapping of stimulus features to reward, so only these models could generalise to unseen feature values (Fig 5B). Hence all learning models were capable of performing our task.

We next evaluated which models could in principle reproduce the above-reported condition difference by simulating the models with a higher learning rate for the slow compared to the fast feature (0.6 vs 0.3, respectively; for the 1LR model, we used $\alpha = 0.3$). As expected, all models with 2 or 4 learning rates (2LR$_f$, 2LR$_c$ and 4LR) could, given appropriate parameters, account for a difference between the slow and fast conditions (Fig 5C), while the 1LR model could not reproduce this effect.

**Learning is affected by slowness.**  Having established that all models in principle represent plausible accounts of behaviour, we next asked which model fits participant choices best, using maximum likelihood fitting and compared models using protected exceedance

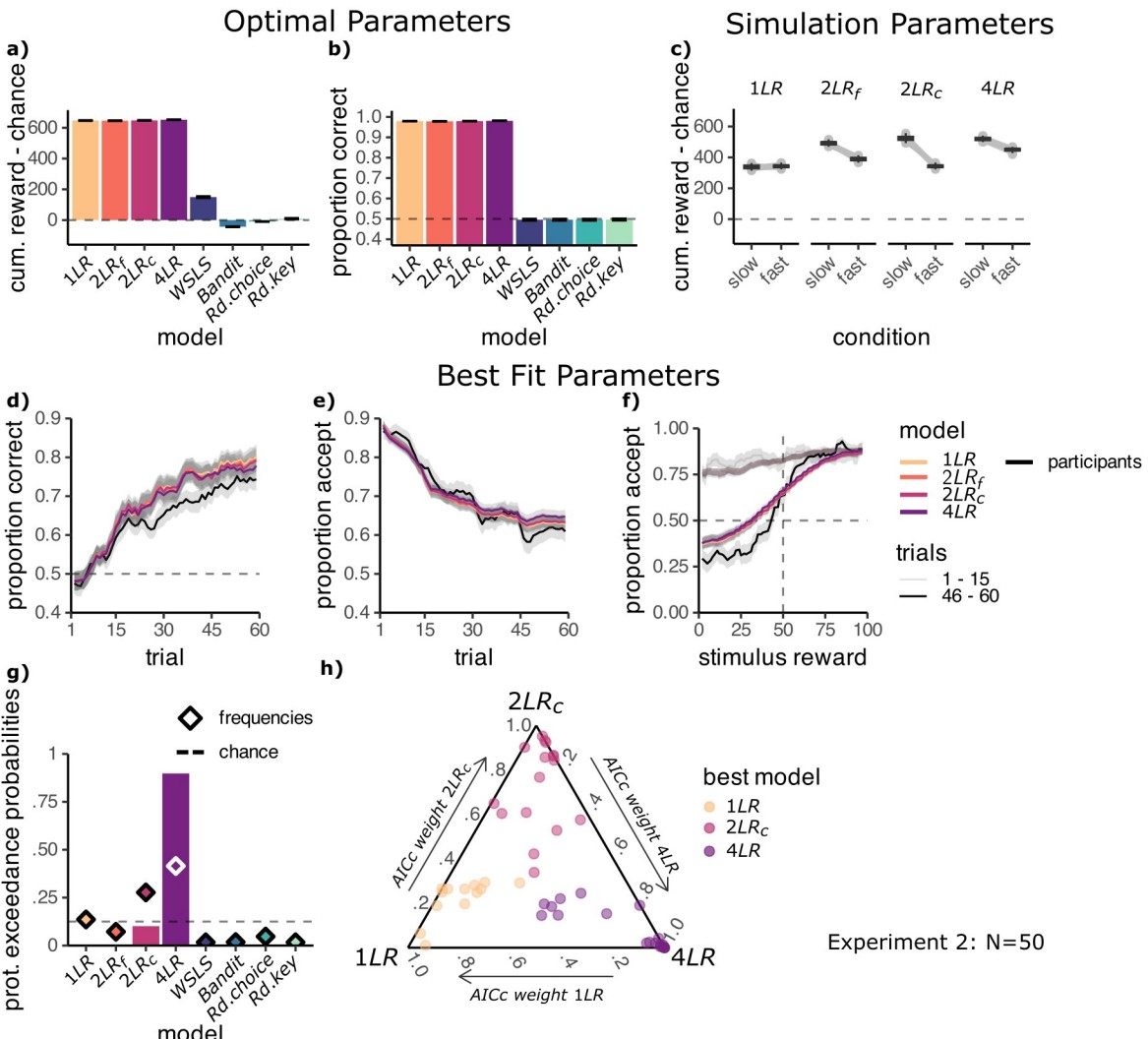

**Fig 5. Models including slowness effect explain participant behaviour best. A:** All learning models can learn the task. Mean reward in the learning phase for the models using reward-maximising parameters. Learning models: one learning rate model (1LR), separate learning rates per feature (2LR$_f$), separate learning rates per condition (2LR$_c$) and the four learning rates model (4LR). Control models: win-stay-lose-shift (WSLS), learning model ignoring features (Bandit), random responding with a bias for accept choices (Rd. Choice) or response key (Rd. Key). **B:** Mean accuracy in the test phase for the models using reward-maximising parameters. **C:** Mean reward for slow and fast blocks in the learning phase for the models simulated using hand-picked learning rates, $\alpha/\alpha_F = 0.3$ $\alpha_S = 0.6$. For the 4LR model both relevant learning rates, $\alpha_{S,R}, \alpha_{F,R}$, were increased by 0.1. **D:** Proportion correct choices across trials in the learning phase. **E:** Proportion of accept choices across trials in the learning phase. **F:** Proportion of accept choices depending on the true stimulus reward, for the first and last 15 trials of the learning phase. **D-F:** Using best fit model parameters. Curves were averaged across 3 adjacent values. Learning models are shown in coloured lines and participants in black. **G:** Protected exceedance probabilities (bars) and estimated frequencies (diamonds) of the models. **H:** Simplex of AICc weights (larger values indicate better fit), calculated considering only the three best-fitting models: 4LR, 2LR$_c$ and 1LR. Each point is one participant, coloured by their best fit model. Plot produced with [52].

probabilities. Protected exceedance probabilities (XP) were calculated with the `bmsR` package in R [53], with model evidence approximated with corrected Akaike Information criterion (AICc) weights, relative to the 1LR model [54], for details see Methods. We first examine the model results in experiment 2 and in a following section validate our findings in the replication experiment. We did not fit the models to the data from experiment 1, as its shorter learning blocks provided less trials for model fitting and it less reliably picked up on the effect of

**Table 1. Best fit parameter estimates for experiment 2.**

|  | $c$ | $\sigma$ | $\kappa$ | $\alpha/\alpha_S/\alpha_{S,R}$ | $\alpha_F/\alpha_{F,R}$ | $\alpha_{S,I}$ | $\alpha_{F,I}$ |
|---|---|---|---|---|---|---|---|
| 1LR | 6.08 (2.84) | 41.96 (20.37) | 6.70 (8.29) | .59 (.33) |  |  |  |
| 2LR$_f$ | 6.57 (3.08) | 44.52 (7.43) | 5.88 (7.43) | .69 (.34) | .55 (.34) |  |  |
| 2LR$_c$ | 6.19 (2.65) | 43.71 (8.76) | 6.82 (8.76) | .61 (.33) | .57 (.32) |  |  |
| 4LR | 6.33 (2.96) | 47.48 (8.01) | 6.80 (8.01) | .78 (.33) | .70 (.37) | .39 (.36) | .40 (.32) |

Mean and standard deviation of the best estimates in experiment 2 for the exploration parameter ($c$), decision noise ($\sigma$), von Mises concentration ($\kappa$), and learning rates on the first trial ($\alpha$) for the slow ($_S$) or fast ($_F$), and relevant ($_R$) or irrelevant ($_I$) feature, obtained through maximum likelihood fitting.

slowness on participant learning. Following maximum likelihood fitting, we first simulated the models with the best-fit parameters (see Table 1). This showed that all models were able to qualitatively match participant learning curves, increasing from 50% to just under 80% correct choices across the 60 trials in a learning block (Fig 5D, see Fig B in S1 Materials for individual participant fits). Models also captured the decrease in accept choices from around 85% to approximately 63% by the end of learning (Fig 5E), as well as the increase in sensitivity to expected reward in the learning phase (Fig 5F).

Notably, in experiment 2, comparing protected exceedance probabilities [55] and AICc scores [56] indicated that the model with four different learning rates (4LR model) fitted behaviour best (XP = .898, AICc = 468.8, see Fig 5G), followed by the model with separate learning rates per condition (2LR$_c$ model, XP = .101, AICc = 470.0) and the 1LR and 2LR$_f$ models (1LR: XP = .001, AICc = 472.7; 2LR$_f$: XP < .001, AICc = 471.7). The 4LR model was estimated as the most frequent model out of those tested (42%), followed by the 2LR$_c$ model (28%, Fig 5G). Together these two models best explained the behaviour of most participants (N = 34/50), however some participants were best fit by the 1LR model (N = 8/50, estimated frequency 14%).

To ask how clear the evidence in favor of the winning model was within each participant, we inspected the distribution of AICc weights for the three best-performing models on a simplex (4LR, 2LR$_c$ and 1LR, Fig 5H). The AICc distribution indicated that participants best fit by the 4LR model were unambiguously best fit by this model, i.e., participants best fit by this model had relatively low weights for the other models. A similar picture emerged for the 2LR$_c$ model. In the case of the one learning rate model (1LR) the difference in fit between the best and alternative models was less pronounced. We fit additional models which allowed parameters other than the learning rate to vary by condition, however these provided a worse account of participant behaviour and are not included in the analyses above (see Fig D in S1 Materials).

In sum, the evidence that the best-performing models, 4LR and 2LR$_c$, adapted their learning rates to the feature speed suggests that participants' learning was affected by feature slowness.

**The 4LR model captures participant behaviour.** Given that the 4LR model emerged as the winning model, we asked how this model related to the behavioural differences between slow and fast blocks in experiment 2. We compared 4LR model fits to the 1LR model to examine the improvement in fit conferred by the adaptation of learning rates to feature speed, while accounting for the remaining learning mechanisms and ability to solve the task, which were the same across all models (see Fig 5A). Simulating 4LR model choices using the best-fit parameters showed a qualitatively similar condition difference in accumulated reward as seen in participants, however this was not significant ($M_S = 319.33 \pm 33.71$, $M_F = 292.51 \pm 31.57$, $t(49) = 1.11$, $p_{1-sided} = .135$, $d = 0.16$, Fig 6A). We found that larger differences in participants' cumulative reward in slow compared to fast blocks in the learning phase were related to a

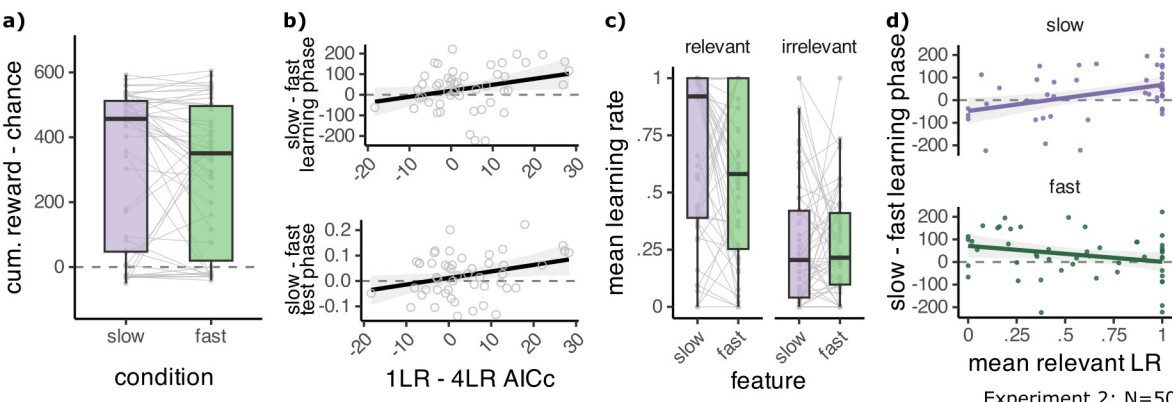

**Fig 6. The four learning rates model captures participant behaviour. A:** Simulating the 4LR model with the best-fit learning rates leads to higher collected reward in slow compared to fast blocks. **B:** A better fit of the 4LR model (x) is related to greater collected reward in slow than in fast blocks in the learning phase (top) and greater accuracy in slow than in fast blocks in the test phase (bottom). **C:** Distribution of learning rates for the 4LR model, obtained from maximum likelihood fitting. Mean across all trials in a block. **D:** Higher mean learning rates for the relevant slow feature (top) are correlated with greater collected reward in slow than in fast blocks in the learning phase (y). Relevant mean learning rates for the fast feature are not correlated with the slowness effect (bottom). Points are individual participants. Line plots are a linear regression line fitted to the data using the least squares method and grey ribbons show the 95% confidence interval.

better fit of the 4LR relative to the 1LR model ($r = .28$, $p = .045$, Fig 6B top). Stronger behavioural effects in the test phase were also related to a better relative fit of the 4LR model ($r = .30$, $p = .032$, Fig 6B bottom). No such relationships were found for the 2LR$_c$ model ($p > .05$, all $p$ values uncorrected).

Further, the fitted learning rates related to participant behaviour. Note that due to the Kalman filter aspect of our model, the learning rates decreased across trials (see Fig D in S1 Materials). Therefore, we examined the mean learning rate across all trials in a block, instead of using the fit value, which was the learning rate on the first trial. When it was relevant, the slow feature benefited from higher mean learning rates than the fast feature ($M_S = .68 \pm .05$, $M_F = .57 \pm .05$, $t(49) = 2.09$, $p = .042$, $d = 0.30$). For the irrelevant learning rates, we found no such difference ($M_S = .28 \pm .04$, $M_F = .27 \pm .03$, $t(49) = 0.16$, $p = .875$, $d = 0.02$, Fig 6C, all $p$ values uncorrected). Larger mean learning rates for the relevant slow feature were correlated with more reward being accrued on slow than on fast blocks in the learning phase ($r = .41$, $p = .012$ Fig 6D top). No other learning rate showed a significant relationship to the behavioural effect (all $p > .05$). These results indicate that the effect of feature speed on learning was mainly modulated by improved learning from the slow feature. Hence, individual differences in model parameters and fit captured differences in how strongly the slowness prior influenced participants' choices.

**The replication confirms 4LR as best model.** To validate our modelling results, we ran the same model fitting procedure and analysis on the learning phase choices of the participants in the replication sample. For best fit parameter estimates see Table 2. The models matched participant behaviour well (see Fig 7A–7C and Fig C in S1 Materials). The results confirmed our previous findings, with an even clearer advantage for the 4LR model relative to the 2LR$_c$, 2LR$_f$ and 1LR models, as indicated by the exceedance probabilities and AICc scores (4LR: XP = 1, AICc = 474.6, N best fit = 67/138, 2LR$_c$: XP < .001, AICc = 477.6, N = 21/138, 2LR$_f$: XP < .001, AICc = 477.6, N = 23/138, 1LR: XP < .001, AICc = 478.8, N = 25/138, Fig 7D). Here, too, the distribution of AICc weights within each participant indicated that most participants best fit by the 4LR model had relatively low weights for the next best models (Fig 7E).

**Table 2. Best fit parameter estimates for the replication.**

|  | $c$ | $\sigma$ | $\kappa$ | $\alpha/\alpha_S/\alpha_{S,R}$ | $\alpha_F/\alpha_{F,R}$ | $\alpha_{S,I}$ | $\alpha_{F,I}$ |
|---|---|---|---|---|---|---|---|
| 1LR | 6.99 (4.23) | 41.04 (17.24) | 4.30 (5.02) | .67 (.25) |  |  |  |
| 2LR$_f$ | 7.15 (4.18) | 42.56 (17.23) | 4.38 (5.28) | .74 (.27) | .64 (.30) |  |  |
| 2LR$_c$ | 7.03 (4.08) | 42.24 (17.25) | 4.73 (5.99) | .69 (.23) | .65 (.28) |  |  |
| 4LR | 6.97 (3.81) | 45.00 (17.49) | 5.33 (6.86) | .88 (.20) | .80 (.27) | .46 (.34) | .43 (.30) |

Mean and standard deviation of the best estimates in the replication for the exploration parameter ($c$), decision noise ($\sigma$), von Mises concentration ($\kappa$), and learning rates on the first trial ($\alpha$) for the slow ($_S$) or fast ($_F$), and relevant ($_R$) or irrelevant ($_I$) feature, obtained through maximum likelihood fitting.

Simulating 4LR model choices using the best fit parameters showed that the model accumulated higher reward in slow than in fast blocks, however this difference was not significant ($M_S = 352.54 \pm 15.90$, $M_F = 331.85 \pm 17.93$, $t(137) = 1.53$, $p_{1-sided} = .065$, $d = 0.13$, Fig 7F). We found that participants better fit by the 4LR model relative to the 1LR model, accumulated more reward in the slow than the fast blocks of the learning phase ($r = .21$, $p = .015$, Fig 7G top). Unlike in experiment 2, this relationship also held for the 2LR$_c$ and 2LR$_f$ models (2LR$_c$: $r = .21$, $p = .015$, 2LR$_f$: $r = .26$, $p = .002$, all $p$ values uncorrected). Similarly, we found that a better fit of the 4LR model relative to the 1LR model was related to better performance in the slow blocks of the test phase ($r = .29$, $p = .001$, Fig 7G bottom). The same relationship was observed for the 2LR$_c$ 2LR$_f$ models (2LR$_c$: $r = .20$, $p = .019$, 2LR$_f$: $r = .27$, $p = .002$, all $p$ values uncorrected).

Again, the slow feature benefited from higher mean learning rates than the fast feature when it was relevant ($M_S = .76 \pm .02$, $M_F = .68 \pm .03$, $t(137) = 2.13$, $p = .035$, $d = 0.18$, Fig 7H). There was no significant difference in the irrelevant mean learning rates ($M_S = .34 \pm .03$, $M_F = .29 \pm .02$, $t(137) = 1.49$, $p = .139$, $d = 0.13$, all $p$ values uncorrected). Additionally, we found that both relevant learning rates were associated with the behavioural effect. More accumulated reward on slow than on fast blocks in the learning phase was linked to larger mean learning rates for the relevant slow feature ($r = .50$, $p < .001$ Fig 7I top) and smaller mean learning rates for the relevant fast feature ($r = -.62$, $p < .001$ Fig 7I bottom). Unlike in experiment 2, this indicates that the effect of slowness on learning was modulated not only by increased learning from the slow feature, but also by decreased learning from the fast feature.

## Discussion

Causal processes tend to evolve on a slower timescale than noise [31]. To investigate whether humans employ a slowness prior to identify potentially relevant features during reinforcement learning, we tested participants in a decision-making task with stimuli composed of one reward-predictive and one reward-irrelevant feature. Participants learned the value of stimuli faster when the reward-predictive feature changed slowly and the irrelevant feature changed quickly, compared to when the opposite was the case. After learning, participants could generalise the learned values to new stimuli equally well in both conditions. By comparing models with different structures for the learning rates, we showed that participants adjusted their learning to the speed of the features. Specifically the learning rates for the relevant features mediated the behavioural effect, suggesting that the observed behavioural differences between conditions were being driven by increased learning from the slow feature and decreased learning from the fast feature. Our study extends research on the slowness prior to humans and suggests that it aids learning task states, in a reinforcement learning domain.

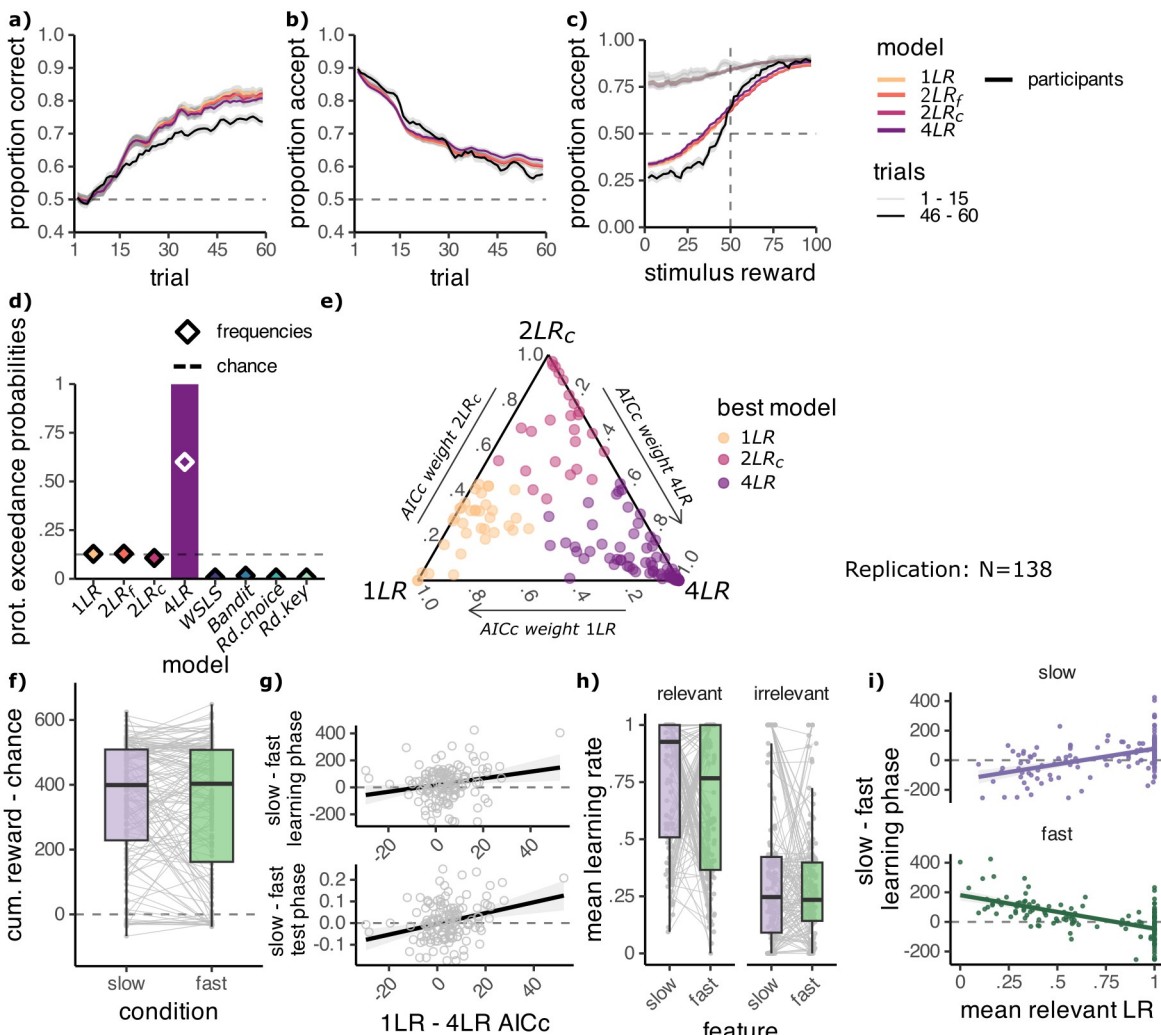

**Fig 7. The replication confirms the four learning rates model captures participant behaviour best. A-C:** Participant behaviour (black) and learning model predictions using best fit parameters showing the proportion of **A:** correct choices and **B:** accept choices across trials in the learning phase and **C:** accept choices depending on the true stimulus reward, for the first and last 15 trials of the learning phase. Lines smoothed across 3 adjacent values. Learning models: one learning rate model (1LR), separate learning rates per feature (2LR_f), separate learning rates per condition (2LR_c) and the four learning rates model (4LR). **D:** Protected exceedance probabilities (bars) and estimated frequencies (diamonds) of the models. **E:** Simplex of AICc weights (larger values indicate better fit), calculated considering only the three best-fitting models: 4LR, 2LR_c and 1LR. Each point is one participant, coloured by their best fit model. **F:** Simulating the 4LR model with the best-fit learning rates leads to higher collected reward in slow compared to fast blocks. **G:** A better fit of the 4LR model (x) is related to greater collected reward in slow than in fast blocks in the learning phase (top) and greater accuracy in slow than in fast blocks in the test phase (bottom). **H:** Distribution of learning rates for the 4LR model, obtained from maximum likelihood fitting. Mean across all trials. **I:** Higher mean learning rates for the slow feature (top) and lower mean learning rates for the fast feature (bottom) are correlated with greater collected reward in slow than in fast blocks in the learning phase (y). Points are individual participants. Line plots are a linear regression line fitted to the data using the least squares method and grey ribbons show the 95% confidence interval.

Our work relates to a broader discussion of how the human brain solves representation learning problems [27, 30]. Previous work has shown how representation learning can be implemented in parallel to reinforcement learning by using feedback signals to guide selective attention [57, 58], or through replay mechanisms during offline periods [59, 60]. Although these approaches represent flexible mechanisms that allow on-the-fly adaptation to the current environment, it is unlikely to be feasible in environments with hundreds of possible signals to

attend to [3, 6, 29]. Our results suggest that for this reason representation learning mechanisms during RL are supplemented with inductive biases. Inductive biases may act by biasing beliefs about which features are relevant, thereby reducing the reliance on online computation. This narrowing down of options could be implemented as selective attention, which recruits frontal, parietal and visual cortical areas [61]. One possibility is that inductive biases modulate activity in these areas to accelerate representation learning. Our findings are in line with previous research showing that priors have a pervasive influence on behaviour, shaping perception [15, 37], remaining stable in the face of exposure to contradictory training [62], and hindering learning of structures which do not align with them [63, 64]. More indirectly, our work raises the question about the origins of such priors, and whether they are learned themselves. One possibility in this regard is that meta-learning, or learning to learn, is the core mechanism that humans use in order to extract regularities of their environment and develop priors that aid perception and learning [65].

While our results align with several theoretical studies on the slowness prior [34, 39, 43], it is important to consider other ways in which slowness can benefit learning. By their nature, slow features are more predictable [35, 36]. This raises the question of whether the learning improvement for slow features is due to them being easier to learn from, rather than being selectively attended to. For instance, the temporal auto-correlation of features and rewards inherent to a slowly changing environment could enable the use of heuristic strategies, such as a win-stay-lose-shift rule [49, 50]. We addressed this concern through model comparison and found that these strategies were unable to explain the behaviour of participants. Another possibility is that presenting stimuli in an ordered fashion yields benefits, as suggested in function learning studies [66]. In our task, slow blocks were more likely to be ordered than fast blocks, but due to the periodic nature of our feature-reward mapping, ordering might not be immediately apparent in either condition. Still, future research should aim to disentangle the effects of ordering and slowness on learning. Importantly, assuming relevant processes change slowly only is a useful assumption given the physical laws that govern our world, i.e., Newton's first law of motion, inertia [39]. Under these conditions, slow acceleration and changes in acceleration are likely to also provide useful priors, as has been shown in motion perception studies in humans [38]. Human learning likely incorporates a host of priors, reflecting other properties determined by our (intuitive) physical understanding of the world [16].

Our findings also relate to previous work on curriculum learning, which has shown that humans benefit from blocked, rather than interleaved, training on a context-dependent categorisation task [67]. In the blocked curriculum the relevant features for categorisation were the same across trials, whereas in the interleaved curriculum the relevant features could switch from trial to trial, even though the stimuli characteristics changed in both curricula. This raises the possibility that slowness, not only in feature dynamics but also in task rules, may aid learning. However, it is worth noting that interleaved training might promote the formation of more generaliseable representations [68], suggesting that the optimal learning curriculum may differ depending on the task at hand. In sum, multiple lines of research point toward a beneficial effect of slowness on learning. Here, we propose that part of this effect is due to the existence of a slowness prior.

Our task and models make some simplifying assumptions. In our task, participants need to reduce a two-dimensional stimulus to a one-dimensional representation. Despite its simplicity, the task itself posed a considerable challenge to participants, as indicated by their end-of-learning performance, which still left room for improvement. Consequently, the task contained the necessary elements to test our hypothesis and provides a controlled test bed for looking at dimensionality reduction. Our winning model, the four learning rate model, assigned learning rates to the features based on their speed and relevance from the first learning trial of the

block. While it is reasonable to assume that participants in experiment 2 and the replication knew the speed of the features based on the preceding observation phase, they could not yet have known which feature was relevant. The models gradually learn the feature relevance through the feature weights, but the structure of the learning rates anticipates the outcome of this learning process. Nonetheless, the fact that a model with this assumption fits best and participants' accuracy increased within the first learning trials in a block, lead us to believe that participants quickly developed a sense for the relevance of the features. We chose this approach for its computational simplicity, but it remains a potential avenue for future research to set up models which do not assign relevant and irrelevant learning rates *a priori*, but infer these assignments through learning. It is for instance possible that the dynamics of learning rates are influenced by a number of additional factors, such as volatility or the size of prediction errors [69–71]. In addition, participants might learn a belief about which feature is relevant to determine learning rates [72].

Overall, the results of our experiments suggest that participants were able to learn the values of stimuli better when the relevant feature changed slowly. By providing empirical evidence for the role of a slowness prior in human learning and connecting to a large number of machine learning findings [31, 39, 41], our study sheds light onto how humans might rapidly learn representations in complex environments.

## Methods

### Ethics statement

All participants provided informed consent in written form and the study was approved by the ethics committee of the Max Planck Institute for Human Development (approval number: N-2020–08).

### Participants

For all experiments participants were recruited through Prolific (www.prolific.co) and completed the experiment online (*experiment 1*: N = 50, female = 19, age = 18–38 years, M = 24.4 years, SD = 5.3 years, *experiment 2*: N = 50, female = 15, age: 18–39 years, M = 24.6 years, SD = 5.4 years, *replication*: N = 195, female = 60, diverse = 2, age: 18–40 years, M = 28.2 years, SD = 5.8 years). None reported being colour blind and none were currently receiving treatment or taking medication for mental illness. Participants were compensated £3.75 (replication £5.00), plus a performance-dependent bonus of up to £1.50 (replication £2.50). To determine the sample size for the replication study, we ran a power analysis based on the test phase effect in experiment 2, as it was the test with the smallest effect size out of the three main tests ($M_S$ = 76% ± 2, $M_F$ = 74% ± 2, $t(49)$ = 1.85, $p_{1-sided}$ = .035, $d$ = 0.26). To achieve ca. 95% power with a paired one-sided t-test, to detect an effect with Cohen's d = 0.26, we aimed for a sample size of 160 participants and collected 195 participants. We introduced a new performance based exclusion criterion in the replication study, to exclude participants with random end-of-learning performance. We defined end-of-learning performance for each participant as the accuracy in the last 15 trials of the learning phase, averaged across all eight task blocks (120 trials). If this mean accuracy was on or below the 97.5th percentile of a binomial distribution over 120 draws with a probability of success of $\frac{1}{2}$ (i.e. 71/120 successes or ca. 59.2% correct), the participant was excluded. After exclusions the effective sample size was 138 participants.

## Materials

Stimuli were coloured shapes, with shapes originating from the Validated Circular Shape space [73] and colours defined as a slice in CIELAB colour space, with luminance 70, chroma 51 and origin [0, 0]. Shapes and colours were parameterized on a circular space, so each position (0–359˚) corresponded to one colour or one shape (Fig 1A), and colour/shape similarity varied continuously but had no hard boundaries. The feature spaces were perceptually uniform, so that the angular distance between feature values corresponded to the perceived difference between them. Small angular distances correspond to similar shapes (or colours), whereas large angular distances correspond to distinct shapes (or colours).

In the learning phase of each block, a subset of 15 positions was shown, spaced uniformly around the circle in steps of 24˚. Each block used a distinct set of positions, offset from the positions used in other blocks in multiples of 3˚ and assigned to blocks in a random order. In the test phase, stimuli were constructed from 15 feature positions offset by 12˚ from the positions used in the preceding learning phase. This offset ensured that shapes and colours seen at test were maximally different from those seen during learning, providing a strong and semi-independent test of participants' knowledge about the feature-reward mapping.

The task was programmed as an online experiment using the jsPsych library version 6.1.0 [74]. The code for the task, including stimuli and instructions can be found on GitHub (https://github.com/noahedrich/slow_prior/tree/main/task).

## Design

Participants completed a task that required them to learn the rewards associated with a set of visual stimuli characterized by two features (colour and shape) (Fig 1). Unbeknownst to participants, stimulus rewards were related to only one of the two features in each block. We refer to the feature that predicted reward as the relevant feature and the feature that did not predict reward as the irrelevant feature (Fig 1B). For each block one position in the relevant feature space was chosen as the maximum reward position. Maximum reward positions were at 10˚, 100˚, 190˚, or 280˚ in the feature space. Each of these reward positions was used once for colour-relevant and once for shape-relevant blocks, in random order. The closer the relevant stimulus feature was to the maximum reward position, the higher the stimulus reward. The stimulus reward was calculated as the absolute distance between the relevant feature position and the maximum reward position, subtracted from the maximum possible distance of 180˚. The resulting value was re-scaled from the angular distance range (0–180˚) to the reward range (0–100 coins).

We manipulated feature speed, by controlling the trial-to-trial variability of the two features. Within each block, one feature had low variability across trials (e.g. participants see relatively similar shapes from trial to trial), while the other feature had high variability (e.g. participants see relatively distinct colours from trial to trial). We refer to these as the slow and fast feature, respectively (Fig 1A). The slow feature was sampled using a Gaussian random walk centred on 0˚, with a standard deviation of 30˚. The fast feature was sampled randomly, while preventing the smallest step-size (24˚) from occurring. Within each block, the 15 feature positions (see Materials) repeated three times in experiment 1 and four times in the experiment 2 and the replication, with each position being shown once before repeating. In this way, we ensured comparable exposure to the slow and fast feature spaces, despite their differing variability.

We counterbalanced the relevant feature dimension (shape relevant/colour irrelevant or vice versa) and the feature speed (shape slow/colour fast or vice versa). Each combination of relevant feature dimension and relevant feature speed was repeated twice, resulting in eight

task blocks. In half of the eight task blocks, the slow feature was relevant (slow blocks), in the other half the fast feature was relevant (fast blocks, Fig 1C). The block order was pseudo-randomised, so that each combination was experienced once before repeating.

## Procedure

Each task block consisted of three phases, observation, learning, and test (Fig 1D–1F).

The observation phase served to demonstrate the variability of the features to participants. Thirty individual stimuli were shown in rapid succession (500ms each) and without intervening screens. The speed of the features in the observation phase matched that in the subsequent learning phase. Both phases used the same set of 15 feature positions, however, sequences for observation and learning were sampled independently and started at randomly selected positions in feature space. In the learning phase, participants played an accept-reject task and were asked to maximise coins earned by collecting valuable gems. Each trial began with a gem (a coloured shape) being displayed centrally on the screen. Using the 'F' or 'J' key, participants could either accept the stimulus, and receive the reward associated with it (between 0 to 100 coins), or reject the stimulus and receive an average reward (50 coins). The reject/accept key mapping was counterbalanced across trials. If participants failed to respond after four seconds they received zero coins. Immediately after a key press, the number of coins earned was displayed on the screen for one second, followed by a blank screen for a variable inter-trial interval (0.5 to 1.5s). A correct response was defined as accepting a stimulus with a value above 50 coins or rejecting a stimulus with a value below 50 coins.

Following the learning phase, participants completed a two alternative forced choice task to test their understanding of the stimulus values. In this test phase, participants were presented with pairs of stimuli and asked to choose the more valuable stimulus in the pair, based on the preceding learning phase. On each trial, participants could choose the left or right stimulus with the 'F' or 'J' keys, respectively, with no time limit. After their response a blank screen was shown for a variable inter-trial interval (0.5 to 1.5s). There was no trial-wise feedback during the test phase. A correct response was defined as choosing the stimulus with the higher value. Here, feature speed was no longer manipulated. Instead, the difference in value between the two stimuli in a pair was systematically varied. By controlling the relevant feature positions of the two stimuli, it was possible to probe choices from easier comparisons, where stimuli had more distinct values (the maximum included difference was 54 coins), to increasingly difficult comparisons, where the values of the two stimuli were more similar(the minimum difference was 13 coins in experiment 1 and 2 coins in experiment 2 and the replication). Overall block accuracy (including both learning and test phase) was reported to participants at the end of the block and used to determine the performance bonus.

We ran two versions of the experiment. In *experiment 1* the observation phase of the experiment was omitted. Nonetheless, the speed of the features was still manipulated during the learning phase, so slowness information was available, but less evident and presented concurrently with the reward learning task. *Experiment 2* included an observation phase prior to the learning phase, as described above, which explicitly demonstrated the speed of the features prior to learning their values. Additionally, there were differences in the length of each task. In *experiment 1* participants completed 45 learning trials and 15 test trials per block, while in *experiment 2* participants completed 30 observation trials, 60 learning trials, and 36 test trials per block. In all other aspects, the experiments were identical. The third collected participant sample was a direct replication of experiment 2, with no changes to the task.

**Mixed effects models.**   We ran logistic mixed effect models in R (R version 4.3.1, RStudio version 2023.09.1 + 494), using `glmer` from the `lme4` package (version 1.1–32) [75–77]. To

obtain parameter values we ran the Bound Optimisation by Quadratic Approximation (BOBYQA) algorithm for 100.000 evaluations. We initially included all relevant fixed effects and their interactions in the models and subsequently used the `drop1` function in R to test which terms contributed to the fit. We corrected all post hoc tests for multiple comparisons using the Bonferroni-Holm method [78] considering the number of terms in the model including the 'condition' variable. All terms that did not significantly improve the fit were removed. We used a maximal random effects structure whenever possible [79]. That is, all variables and interactions initially included as fixed effects were included in the random effects, even if they were later dropped from the fixed effects. Random effects were only simplified if the maximal structure led to fitting issues. All continuous predicting variables were scaled, trial number was normalised to range between zero and one. Trials with no response were excluded from all analyses.

To examine the effect of reward on choices in the learning phase we used a logistic mixed effects model to predict choices based on condition (slow/fast), the stimulus reward on each trial, trial number and all two-way interactions. The examined model was:

$$
\begin{aligned}
C_t \quad &= \beta_0 + \beta_1\, \text{Condition}_t + \beta_2\, R_t + \beta_3\, t \\
&+ \beta_4\, \text{Condition}_t \times R_t + \beta_5\, \text{Condition}_t \times t + \beta_6\, R_t \times t \\
&+ (1 + \text{Condition}_t + R_t + t | \text{Subject})
\end{aligned}
\tag{6}
$$

where $C_t$ is the participant choice on trial $t$ (0 = reject, 1 = accept), and the predictors are the Condition (slow/fast block), stimulus reward $R_t$, the trial number $t$.

To examine the effect of the relevant and irrelevant feature on choice we used a logistic mixed effects model to predict choices based on the stimulus colour and shape positions on each trial. As the features were angles in the shape and colour circles, each feature was included as a `cos()` and `sin()` predictor in the model. This analysis was run separately for slow and fast blocks.

$$
\begin{aligned}
C_t \quad &= \beta_0 + \beta_1\, t + \beta_2\, cos(\theta_R) + \beta_3\, sin(\theta_R) + \beta_4\, cos(\theta_I) + \beta_5\, sin(\theta_I) \\
&+ \beta_6\, t \times cos(\theta_R) + \beta_7\, t \times sin(\theta_R) + \beta_8\, t \times cos(\theta_I) + \beta_9\, t \times sin(\theta_I) \\
&+ (1 | \text{Subject})
\end{aligned}
\tag{7}
$$

where $\theta_R$ is the position of the relevant feature and $\theta_I$ is the position of the irrelevant feature. For the replication study it was possible to fit the following more extensive random effects structure:

$$
(1 + t + cos(\theta_R) + sin(\theta_R) + cos(\theta_I) + sin(\theta_I) | \text{Subject})
$$

To assess whether the irrelevant feature impacted choices, we compared the full model to one without the `cos()` and `sin()` predictors for the irrelevant feature. Note that the preregistration states that we do not run model comparison for this analysis. We do not exclude predictors based on model comparison, but we do use it to assess the contribution of predictors.

To examine the effect of reward on choices in the test phase we used a logistic mixed effects model to predict choices based on the condition (slow/fast), difference in reward between the stimuli on each trial, and their interaction:

$$
C_t = \beta_0 + \beta_1 \text{Condition}_t + \beta_2 R_{\text{diff},t} + \beta_3 \text{Condition}_t \times R_{\text{diff},t} + (1 | \text{Subject})
\tag{8}
$$

where $R_{\text{diff},t}$ is the difference in value between the left and right stimulus on trial $t$. For

experiment 2 it was possible to fit the following random effects:

$$(1 + \text{Condition}_t | \text{Subject})$$

For the replication it was possible to fit the full random effects:

$$(1 + \text{Condition}_t + R_{\text{diff},t} + \text{Condition}_t \times R_{\text{diff},t} | \text{Subject})$$

**Meta-analysis.** To assess the joint evidence across experiments, we conducted a meta-analysis using the `meta` package in R (version 7.0–0) [80], with all statistics calculated as in RevMan 5 [81]. We assumed a fixed effect model and used Hedge's g to assess the effect size. Between-study heterogeneity was assessed using Cochran's Q test.

**Plotting.** All figures were generated with the `ggplot2` and `ggtern` packages in R [52, 82]. Colours for the models were sourced from `viridis` [83]. Post-processing was done in Inkscape [84].

## Computational models

To analyse trial-by-trial learning, we fit ten computational models to the choices of participants in the learning task. Four learning models embodied alternative hypotheses about how the prior could affect learning and differed in their ability to adapt their learning rates to the slowness of the features. The other six models, four of which are reported in the main text, served as control models and tested for competing hypotheses or tested whether participants engaged with the task. All model code is available on GitHub at https://github.com/noahedrich/slow_prior/tree/main/code/models.

**Learning models.** The reinforcement learning (RL) models used the outcome of each trial to update their estimate of the value of the features and predict the next choices of participants. To account for the fact that continuous feature dimensions in the task allowed participants to generalise their learning within each feature (i.e., learning about the value of red was also informative of the value of orange), stimuli were represented as a distribution in feature space, instead of being represented as only their specific colour and shape angles (Fig 4). A stimulus on trial $t$ was represented as a feature vector $\mathbf{x}_t$. Note that, as each stimulus was made up of two feature dimensions, it was represented by two feature vectors: one for the slow, $\mathbf{x}_{t,S}$, and one for fast-changing feature, $\mathbf{x}_{t,F}$ (corresponding to colour/shape as determined by the current block condition). Therefore, the feature vector for a stimulus $\mathbf{x}_i$ was the concatenation of the slow and fast feature vectors: $\mathbf{x}_t = [\mathbf{x}_{t,S}, \mathbf{x}_{t,F}]$. The feature vectors for the slow and fast feature angles of a stimulus were obtained from a von Mises like distribution, which approximates a normal distribution in circular space, as follows:

$$x_{t,i} = \frac{e^{cos(d_{t,i})\kappa}}{\sum_{i=1}^{360} e^{cos(d_{t,i})\kappa}} \tag{9}$$

where:

$$d_{t,i} = \frac{\theta_t - \theta_i}{360} 2\pi \tag{10}$$

where $x_{t,i}$ is the $i$th entry of feature vector $\mathbf{x}_t$, and $d_{t,i}$ is the distance from the stimulus' feature angle on trial $t$ to feature angle $i$. The parameter $\kappa$ determines the concentration of the function. With large $\kappa$, the distribution becomes concentrated around the stimulus feature angle, and less surrounding angles are included. With $\kappa$ approaching 0, the distribution becomes

uniform. Representing stimuli in this way allowed the model to learn about the value of unobserved angles, based on perceptual similarity.

For each of the two feature dimensions, the models learned a feature weight vector, $\mathbf{w}_{t,S}$ and $\mathbf{w}_{t,F}$, which were concatenated in the weight vector $\mathbf{w}_t = [\mathbf{w}_{t,S}, \mathbf{w}_{t,F}]$. This vector corresponds to the estimated value for each feature position on trial $t$. The expected value $V_t$ of a stimulus on trial $t$ was calculated as the inner product of the feature vector $\mathbf{x}_t$ with the weight vector $\mathbf{w}_t$:

$$V_t = \mathbf{x}_t^T \mathbf{w}_t \tag{11}$$

This value estimate flowed into the prediction of the choice on the next trial and could guide choices to maximise reward. However, before being fully guided by value estimates, it is necessary to gather information and become certain that the estimates are meaningful (as participants do, see Fig 2B). To mediate between the pressures of exploring and exploiting, we supplemented the value estimate for each stimulus with an exploration bonus $U_t$, which reflects how uncertain the model is in its value estimate. The value of accepting stimulus on trial $t$, $V_{a,t}$, was then calculated as follows:

$$V_{a,t} = V_t + c \cdot U_t \tag{12}$$

where $c$ mediates how strongly the exploration bonus is weighted at choice.

Due to the continuous nature of the features and the flexible recombination of features across stimuli, a simple count-based uncertainty estimate (as in the Upper Confidence Bound method [51]) would be ineffective. Instead, specifying the models as Kalman Filters allowed us to take a rigorous approach to estimating the uncertainty on each trial. In addition to tracking a mean value, Kalman Filters keep an estimate of the variance around that mean, which embodies the uncertainty inherent to the estimate. Similar to the feature and weight vectors, the variance estimates were saved in a variance vector $\mathbf{v}_t$, which was a concatenation of slow and fast variance vectors: $\mathbf{v}_t = [\mathbf{v}_{t,S}, \mathbf{v}_{t,F}]$. The exploration bonus was the inner product of the feature vector with the variance vector:

$$U_t = \mathbf{x}_t^T \cdot \mathbf{v}_t \tag{13}$$

While the features shown on each trial changed, the mapping between the feature and the reward was stationary within each block. Therefore, the uncertainty was highest at the beginning of each block and steadily reduced with each observed outcome.

When predicting the next choice, the models compared the value of accepting $V_{a,t}$ with the value of a rejecting, by testing for the probability of $V_{a,t}$ under a cumulative normal distribution centred on 50, with a standard deviation $\sigma$:

$$\begin{aligned} p(\text{accept}) &= P[X \le V_{a,t}] \\ X &\sim N(50, \sigma^2) \end{aligned} \tag{14}$$

Here a smaller $\sigma$ means a steeper increase in accept probability with increasing $V_{a,t}$.

After an 'accept' choice the reward outcome $R_t$ of the trial $t$ is used to update the value and uncertainty estimates. The reward prediction error is used to update weight vector with a learning rate $\alpha_t$, as follows:

$$\mathbf{w}_{t+1} = \mathbf{w}_t + \alpha_t \, \mathbf{x}_t \, (R_t - V_t) \tag{15}$$

The variance vector is reduced by an amount proportional to the learning rate $\alpha_t$:

$$\mathbf{v}_{t+1} = \mathbf{v}_t - \alpha_t \, \mathbf{x}_t \, \mathbf{v}_t \tag{16}$$

Finally, the Kalman Filters also update the learning rate on each trial, as with decreasing uncertainty about the value estimates, smaller updates to the weight vector are needed.

$$\alpha_{t+1} = \frac{U_t}{U_t + M} \tag{17}$$

where $M$ is the constant measurement noise.

All four learning models included the three free parameters, $\kappa$, $c$ and $\sigma$, as specified in the equations above, but they differed in their ability to adapt their learning rates to the slowness of the features (Fig 4). A one learning rate (1LR) model used the same learning rate $\alpha$ regardless of feature speed and thus was indifferent to feature variability and could not account for a difference in performance between the slow and fast blocks. A two learning rates model sensitive to feature variability (2LR$_f$) used different learning rates for the slow $\alpha_S$ and fast $\alpha_F$ changing feature across all blocks, irrespective of whether they were relevant or irrelevant. Another two learning rates model, this one sensitive to block condition (2LR$_c$), used different learning rates, depending on whether the relevant feature was changing slowly $\alpha_S$ or quickly $\alpha_F$ (but used the same learning rate for both features within the block). Finally, a four learning rates (4LR) model had learning rates sensitive to both the feature variability and the block condition. Meaning it had separate learning rates for the slow and fast-changing features when they were relevant ($\alpha_{S,R}$, $\alpha_{F,R}$) and irrelevant ($\alpha_{S,I}$, $\alpha_{F,I}$).

In models with separate learning rates for the slow and fast feature (2LR$_f$ and 4LR), the uncertainty $U_t$ (Eq 13) and learning rates $\alpha$ (Eq 17) were calculated separately for the slow $\mathbf{x}_{t,S}$ and fast $\mathbf{x}_{t,F}$ feature vector. Accordingly, the weight and variance vectors for the slow and fast features were updated with their respective learning rates. To keep comparable magnitudes of learning rates between models, in models with the same learning rate for both features in a block (1LR and 2LR$_c$), we calculated the uncertainty separately for the slow and fast feature and used their mean to update the learning rate according to Eq 17.

**Control models.** We implemented a control model with the same Kalman Filter machinery, which treated the task as a single, stationary bandit for which it estimated a mean and variance (Bandit model). By ignoring the stimulus features, this model could only learn from the reward outcomes. This model was critical to rule out that learning might be easier on slow blocks, simply due to the reward on the current trial being more predictive of the reward on the next trial, irrespective of the variability of the features. Equations were similar to the models of interest, obviating the need for vectors. A single value $V$ and uncertainty $U$ estimate were kept. These were combined as in Eq 12 to the value of accepting $V_a$ with the mediating parameter $c$. The same choice rule as in Eq 14 was used. The value and uncertainty estimates, and the learning rate were updated according to:

$$V_{t+1} = V_t + \alpha_t \left( R_t - V_i \right) \tag{18}$$

$$U_{t+1} = U_t - \alpha_t U_t \tag{19}$$

$$\alpha_{t+1} = \frac{U_t}{U_t + M} \tag{20}$$

where $M$ is the constant measurement error.

To account for a choice perseverance strategy, which could selectively benefit performance in slow blocks where the correct choice on the previous trial was likely the same as the correct choice on the current trial, we included a win-stay-lose-shift model (WSLS model). When the choice on the previous trial was 'accept' and the received reward was equal to or above the

default value of 50, this was counted as a win and the model was likely to choose 'accept' again. In contrast, if the outcome of an 'accept' choice lay below 50, this was counted as a loss and the model was likely to choose 'reject' on the next trial. In both cases the model could instead make the less likely choice with probability $\epsilon$. As 'reject' choices always resulted in a reward of exactly 50 no wins or losses as such were possible, so the model continued to make 'reject' choices and switched to 'accept' with probability $\epsilon$. The first choice was made randomly. The WSLS model can be described as follows:

$$p(\text{accept}) = \begin{cases} 1 - \epsilon, & \text{if choice}_{t-1} = \text{ accept and } R_{t-1} \geq 50. \\ \epsilon, & \text{otherwise.} \end{cases} \tag{21}$$

In addition, we set up models which did not learn and responded randomly, with either a bias to 'accept' or 'reject' (Random Choice model), with choices given by:

$$p(\text{accept}) = b_a \tag{22}$$

or a bias for the left or right response key (Random Key model), with choices given by:

$$p(\text{accept}) = \begin{cases} b_r, & \text{if right key is 'accept'.} \\ 1 - b_r, & \text{otherwise.} \end{cases} \tag{23}$$

To test whether parameters other than the learning rate could explain differences in participant behaviour between the slow and fast blocks, we fit two additional learning models. The first model allowed the the exploration parameter $c$, which mediates the extent to which uncertainty increases the value of accepting (see Eq 12), to be different for slow and fast blocks (1LR$_c$). The second model allowed the decision noise parameter $\sigma^2$ to vary by condition (1LR$_{\sigma^2}$, see Eq 14). Otherwise, both were identical to the one learning rate model (1LR). The results of these models are only included in the supplementary information in Fig D in S1 Materials.

**Model fitting.**   Models were fit to each participant's data in the training trials using the `nloptr` package version 2.0.3 in R [85] by minimising the negative log likelihood with the `NLOPT_GN_DIRECT_L` optimisation function run for 10.000 evaluations. We initialised the learning models and the Bandit model, so that on the first trial of each block, the value estimate of the stimulus $V_t$ was 50 (the same as the value of rejecting), and the uncertainty bonus $U_t$ was 5 for each feature. At the start of fitting, the measurement error $M$ was adjusted so that the learning rate $\alpha_t$ on the first trial would be equal to the fit learning rate (Eqs 17 and 20).

We quantified the reliability of parameter estimates through parameter recovery for the learning rates of the learning models (see Fig E in S1 Materials). The fitting procedure provided fair to excellent reliability, with a high correspondence between ground truth and recovered learning rates.

**Model comparison.**   We simulated model choices given the parameter values obtained from maximum likelihood fitting and obtained the predicted likelihoods for participant choices. These likelihoods were used to calculate the Akaike Information, corrected for small samples [86]:

$$\text{AICc} = 2k - 2LL + \frac{2k(k+1)}{N-k-1}$$

Where k is the number of free parameters of the model, LL is the log likelihood of the data given the model and fit parameters and N is the sample size (i.e. number of trials used to obtain the LL).

We then calculated AICc weights, which provide a measure of goodness of fit of a model relative to a baseline model (for which we chose the 1LR model) [54], as follows:

$$\text{AICc weight} = \frac{e^{-\frac{1}{2}\Delta\text{AICc}}}{\sum_{m\in M}e^{-\frac{1}{2}\Delta\text{AICc}_m}} \tag{24}$$

where $\Delta$AICc is the difference in AICc between the AICc of the model and the baseline model, and $M$ is the set of all models $m$. AICc weights are normalised to sum to one for each participant, with larger values indicating a better fit. Finally, we used AICc weights as an approximation of model evidence to calculate protected exceedance probabilities with the `bmsR` package in R [53, 55].

We tested model identifiability through model recovery, using the same fitting and model comparison procedure as for participants (see Fig E in S1 Materials). Model recovery proved to be reliable, identifying the model that had generated the data correctly for most simulations. The 1LR$_{\sigma^2}$ model had poor recoverability. The 4LR model was sometimes misidentified as the 2LR$_c$ and to a lesser extent the 2LR$_f$ model, likely because when two of the four 4LR learning rates are similar to each other, this model can be identical to the 2LR models.

## Supporting information

**S1 Materials. Supporting materials.** Supplementary information file including additional information on the preregistration, participant behaviour, mixed effect model results, model fits and analyses of additional control models and parameter and model recovery. This file includes Text A (Preregistration), Fig A (Participant learning curves and meta-analysis), Tables A-N (Effect sizes and confidence intervals for the mixed effects models), Fig B (Individual participant and best fit model learning curves for experiment 2), Fig C (Individual participant and best fit model learning curves for the replication), Fig D (Model performance) and Fig E (Parameter and model recovery). Figure legends see inside S1 Materials.
(PDF)

## Acknowledgments

The authors would like to thank Ondrej Zika and Christoph Koch for helpful discussions on the models and Anika Löwe and Georgy Antonov for their valuable feedback on the manuscript.

## Author Contributions

**Conceptualization:** Noa L. Hedrich, Eric Schulz, Sam Hall-McMaster, Nicolas W. Schuck.

**Data curation:** Noa L. Hedrich.

**Formal analysis:** Noa L. Hedrich, Sam Hall-McMaster, Nicolas W. Schuck.

**Funding acquisition:** Nicolas W. Schuck.

**Investigation:** Noa L. Hedrich.

**Methodology:** Noa L. Hedrich, Eric Schulz, Sam Hall-McMaster, Nicolas W. Schuck.

**Project administration:** Nicolas W. Schuck.

**Resources:** Nicolas W. Schuck.

**Software:** Noa L. Hedrich.

**Supervision:** Eric Schulz, Sam Hall-McMaster, Nicolas W. Schuck.

**Validation:** Noa L. Hedrich, Nicolas W. Schuck.

**Visualization:** Noa L. Hedrich, Nicolas W. Schuck.

**Writing – original draft:** Noa L. Hedrich, Nicolas W. Schuck.

**Writing – review & editing:** Noa L. Hedrich, Eric Schulz, Sam Hall-McMaster, Nicolas W. Schuck.

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
