## [Decision Letter · Decision Letter 0]

13 Mar 2024

Dear Dr Schuck

Thank you very much for submitting your manuscript "An inductive bias for slowly changing features in human reinforcement learning" for consideration at PLOS Computational Biology.

As with all papers reviewed by the journal, your manuscript was reviewed by members of the editorial board and by several independent reviewers. In light of the reviews (below this email), we would like to invite the resubmission of a significantly-revised version that takes into account the reviewers' comments.

As you can see all the reviewers found your paper addressing an important question, however the opinions were mixed concerning the robustness and validity of the statistical support of the hypotheses (marginal significant test, seemingly skewed distribution of p-values). Some lack of clarity concerning the relation between the main experiment and pilot, on one side, and deviations from preregistered analyses, on the other side, also seem to undermine the reader's confidence on the results. On this ground, and in order to allow you presenting a paper that unquestionably supports your interesting claims, I would welcome a major-revised manuscript that includes an additional (third) experiment where you assess the (direct) replication of the main findings, thus removing any residual doubt about the statistical and procedural validity of the study. Since the experiments were performed online and the analytical codes are already there, I expect that this should feasible within the delay, but we will of course grant extra time for revising the manuscript and adding the third experiment, if necessary.

We cannot make any decision about publication until we have seen the revised manuscript and your response to the reviewers' comments. Your revised manuscript is also likely to be sent to reviewers for further evaluation.

Sincerely,

Stefano Palminteri

Academic Editor

PLOS Computational Biology

Thomas Serre

Section Editor

PLOS Computational Biology

As you can see all the reviewers found your paper addressing an important question, however the opinion were mixed concerning whether the statistical support to the hypothesis is robust enough (marginal significant test, seemingly skewed distribution of p-values) . Some lack of clarity concerning the relation between the main experiment and pilote and deviations from preregistered analyses also seem to undermine the confidence on the results. On these ground, and in order to allow you presenting a paper that unquestionably support your interesting claims, I would welcome a major-revised manuscript that includes an additional (third) experiment where you assess the (direct) replication of the main findings, thus removing any residual doubt about the statistical and procedural validity of the study. Since the experiments were performed online and the analytical codes are already there, I expect that this should feasible within the delay, but we will of course grant extra time for revising the manuscript, if necessary.

Reviewer's Responses to Questions

**Comments to the Authors:**

Reviewer #1: Hedrich and colleagues investigate feature learning in a two-dimensional bandit learning task where one of the features changed quickly, while the other changed slowly. The authors report that people pay more attention to features that change slowly, and have a bias in favor of those features even when they are not predictive of rewards. Using formal modeling, the authors found that four separate learning rates are required to properly capture participants' behavior -- paying attention to fast- and slow-changing features in relevant and irrelevant conditions separately.

The good features of the paper include the very good presentation (clear writing, informative and pretty plots) and the interesting topic. I particularly liked the task, it is very clever and well suited to investigate the topic of interest. The only thing that I would change, and I strongly urge the authors to do that, is to remove the choice--information coupling during learning. I do not see any reason why outcome feedback should be coupled to choices, as the main interest lies in feature learning and not in exploration/exploitation. In the present state, there is a strong confound of the potential to learn features and behavioral tendencies such as ambiguity aversion. The models seem decently motivated and well executed, I have few concerns in that aspect.

With all that said, what was submitted as a finished paper serves at most as a registered report. As far as I can tell, the only behavioral result that stands up to rigor is that, on average, participants tended to distinguish good options from bad ones. All other analyses are either misrepresentations of various degrees or simply wrong.

First, the preregistration, the data, and analyses. None of these are publicly available, so I could not check what is contained therein and what is not. The authors are transparent about deviations from the preregistration (which is good), but ignoring preregistered exclusion criteria that would remove more than 50% of the sample is bad. The authors claim that "[t]he main results remained unchanged when applying this exclusion criterion" which is plain wrong. In fact, only one of the four results reported after exclusion is statistically significant, and even this one result is probably overly optimistic (see third point below).

Second, and related, is the overuse of "marginal significance" while operating in the standard NHST framework. I counted at least six such occurrences, and I think this blog post summarizes the problem here the best: https://mchankins.wordpress.com/2013/04/21/still-not-significant-2/

Third, almost all p values of significant effects are just below .05 using one-sided tests. Most of them would disappear using two-sided tests, and especially so with any reasonable correction for multiple comparisons. There are good resources why this is problematic, for example, https://www.the100.ci/2018/02/15/the-uncanny-mountain-p-values-between-01-and-10-are-still-a-problem/ and https://quentinandre.net/post/large-pvalues-and-power-analysis/

Fourth, the way the two studies are presented is inconsistent with best practices. It seems like not only the analyses, but also the experimental design deviate from the preregistration, which is very odd. Generally speaking, while I would consider reporting a pilot study in the main text and using it to draw inferences acceptable (but not necessarily good practice), I do not understand why the power analysis was conducted using yet another study that is not reported, and why it has not been adjusted after observing the noise in the pilot study. Generally speaking, in line with the first comment on data exclusions, both the pilot and main studies were severely underpowered.

Fifth, statements about changes in effects are made without supporting them with tests. For example: "however the effect in test phase accuracy was stronger than with the full sample", or "Performance during the test phase did not differ statistically from end-of-learning performance in the learning phase [...]. Hence, our data suggests that participants generalised values successfully across task and stimulus differences between the two phases.". See http://www.stat.columbia.edu/~gelman/research/published/signif4.pdf on why these claims are wrong.

Reviewer #2: In this paper, the authors test a novel hypothesis of how humans might learn useful priors for task representation, namely whether people are biased to learn more from slowly-changing features in the environment. The experimental task is well designed, the paper clearly written, and the set of questions interesting and under-explored in computational cognitive neuroscience. That said, the conclusions we can draw from the paper, and particularly the computational modeling analysis, could be stronger. Below are some comments that I hope will help sharpen the paper.

(1) The authors mention running two separate experiments, but it was not clear whether the data shown here is collapsed across the two (e.g. does Figure 2 show data from all 100 participants?); it would be important to know if the presence of an observation phase changes the conclusions of the study.

(2) Upon further reading, the pilot experiment was analyzed separately and included in the supplement: please make this clear in the subheadings earlier in the paper, and by adding the sample size to the plots in Figure 2.

(3) Did the performance difference between conditions emerge gradually? That can be addressed by plotting 2a as a function of condition and including trial as a variable. We would expect that to be the case based on 2f, since correct acceptance should increase cumulative reward.

(4) If we zoom in on Figure 2F, first 10 trials, it appears that the pattern of results is flipped, such that there is an initial decrease in cumulative reward for the "slow" condition. Seeing as fast-changing features are initially more likely to capture attention (https://jov.arvojournals.org/article.aspx?articleid=2750470), it would be interesting to understand how this effect interacts with the slow feature bias.

(5) The lack of a condition difference in the pilot experiment suggests a top-down mechanism by which people select which features to pay attention to. What could this mechanism be? One can imagine different explanations (e.g. reducing visual working memory load), it would be worthwhile to consider some in the discussion.

(6) I was a bit confused by the interpretation of the computational models. As I understand it, the winning model allows for LR differences based on condition (slow vs. fast). That seems reasonable, as people already know which feature changes slowly vs. quickly from the observation phase. However, allowing the model to also know the relevant feature in advance is peeking, since people have to learn it from trial and error. So the 2LR_c (relevant feature changing fast and slow) model and 4LR model (both relevant and irrelevant feature changing fast and slow) seem like oracle models that, unsurprisingly given number of trials, likely recapitulate the (asymptotic?) learning effect without modeling it directly. Can the authors clarify the choice of modeling approach?

(7) More broadly, the interesting question the authors set out to answer is, do people have a bias to learn from slowly-changing features first? The paper provides evidence for this in the form of a behavioral effect, but this conclusion is less warranted by the computational analysis. To show that, you would need a model that explicitly infers feature relevance and biases that inference, or at least early exploration, in favor of slow-changing features.

Minor:

- Figure 3: please explain acronyms (e.g. 2LRc and 2LRf) in the figure caption

Reviewer #3: Thank you for the opportunity to review this manuscript entitled, “An inductive bias for slowly changing features in human reinforcement learning.” In this paper, the authors tested the hypothesis that humans learn better from reward-predictive features when those features change slowly over time. Moreover, this learning bias is assumed to result from people’s prior knowledge that, in most environments, signal tends to change on a slower timescale than noise. Participants completed a bandit task in which the options had two features (color/shape), one changing slowly and the other changing quickly within each block of trials. Moreover, one of the features (slow or fast) was predictive of reward, while the other was not. Participants obtained greater cumulative reward and achieved better generalization performance when the reward-predictive feature changed slowly and the irrelevant feature changed quickly. Reinforcement learning (RL) models suggested that learning rates were higher for slow features, but only when those features were relevant to the task (i.e., predictive of reward).

Overall, I thought the manuscript was well-written, the methods appropriately chosen, and the analyses carefully and correctly executed. I think the research question is interesting and the results were generally supportive of the hypotheses. I have a few questions, comments, and suggestions that I hope will be constructive to the authors.

1. As I was reading pp. 2-3, the immediate question I had was whether people are actually biased toward focusing on slowly changing features, or if it is simply easier to detect/predict them? I would appreciate if the authors could discuss in greater detail whether their results can help answer this question.

2. I suggest reporting somewhere in the main text the number of individual participants who showed significantly better learning in slow compared to fast blocks (i.e., in both cumulative reward earned and generalization performance).

3. The computational modeling analysis assumes that the effects of slow vs. fast changing features are mediated by the learning rate parameter. This seems plausible, but have the authors tested alternative possibilities? For example, could the results be explained equally well if a different parameter (e.g., the noise parameter, sigma^2) varied across conditions?

4. In the Kalman filter-based models considered here, learning rates decrease across trials. How dependent are the conclusions on this particular choice of model? Would the results hold if learning rates were constant across trials?

5. Line 195: should this be Fig. 2f?

6. Line 217: should this be Fig. 2g?

7. Line 839: should be either maximizing LL or minimizing negative LL.

**Have the authors made all data and (if applicable) computational code underlying the findings in their manuscript fully available?**

Reviewer #1: **No: **Neither data nor code available at the time of review

Reviewer #2: None

Reviewer #3: None

PLOS authors have the option to publish the peer review history of their article (what does this mean?). If published, this will include your full peer review and any attached files.

Reviewer #1: No

Reviewer #2: No

Reviewer #3: No
---

## [Decision Letter · Decision Letter 1]

17 Oct 2024

Dear Dr Schuck,

We are pleased to inform you that your manuscript 'An inductive bias for slowly changing features in human reinforcement learning' has been provisionally accepted for publication in PLOS Computational Biology.

Best regards,

Stefano Palminteri

Academic Editor

PLOS Computational Biology

Thomas Serre

Section Editor

PLOS Computational Biology

Reviewer's Responses to Questions

**Comments to the Authors:**

Reviewer #1: This is my review of the revised paper by Hedrich and colleagues, after having reviewed the initial submission. In my original review, I was quite enthusiastic about the topic and its relevance and the good presentation but I had major concerns about meta-scientific aspects of the manuscript, as the authors severely deviated from the preregistration (comment #1), misused statistical significance terms (comments #2 and #5), reported p values just below .05 using one-sided tests (comment #3), and reported a main experiment that was severely underpowered despite having access to pilot data (comment #4). In short, I had concerns about the evidential value of the presented data and analyses. The authors' approach to addressing these concerns was to run a preregistered replication study of the main experiment.

After carefully going through the response letter, the revised manuscript, checking the preregistration, taking a look at the raw data, and the analysis code, I'm afraid my general assessment of the evidential value of the presented paper has not increased. I already expressed my concerns with what is now called "Experiment 1" and "Experiment 2", so I will not repeat these concerns though I will note that their results have gained prominence by being featured as the main experiments in the paper. I will express my concerns with what the authors call "replication".

First is the question of correction for multiple comparisons. In the preregistration, the authors explicitly preregister no such correction because the three main analyses rely on different dependent variables. In the response letter, they justify the lack of corrections because of correlations in the dependent variables. Neither of these two arguments is valid. If the authors reported analyses using 1000 different dependent variables and found 50 significant results, would they have confidence in the authenticity of these findings? The correlation between different dependent variables indeed reduces an inflation of type-1 errors, but unless they are perfectly correlated, there is still an inflation. Multivariate models provide a practical instrument to make such corrections for multiple comparisons. Given the - yet again - extreme density of p values just below .05, even for one-sided test, the only result that might be robust is that participants are able to accumulate more points with slowly changing features.

Second is the choice of model comparison metric. A visual inspection of all figures reveals that the models' absolute fits barely differ, and their relative fits are numerically very close to each other. I was unable to run the R script without encountering several issues and due to the unequal number of trials across participants, I was unable to compute the BICs, but I am fairly certain that a BIC analysis would favor the simplest 1LR model. AIC-based model selection metrics are known to favor complex models over simple models, so if the authors want to make a case for the most complex model, they need to use a more conservative metric.

These two points illustrate my lack of confidence in the evidential value of the reported results. I am far from convinced that there is anything going on apart from the fact that people seem to be able to identify when an option is very good and when it is very bad. None of the analyses related to the difference between quickly changing features and slowly changing features provides convincing evidence that these conditions differ (but they also do not provide convincing evidence that they do NOT differ).

Reviewer #2: I thank the authors for their thorough analysis in response to my comments, and have no further feedback.

Reviewer #3: My concerns with the original manuscript were all addressed. I thank the authors for their thorough responses.

**Have the authors made all data and (if applicable) computational code underlying the findings in their manuscript fully available?**

Reviewer #1: Yes

Reviewer #2: None

Reviewer #3: None

PLOS authors have the option to publish the peer review history of their article (what does this mean?). If published, this will include your full peer review and any attached files.

Reviewer #1: No

Reviewer #2: No

Reviewer #3: No

---

## [Editor Report · Acceptance letter]

18 Nov 2024

PCOMPBIOL-D-24-00120R1 

An inductive bias for slowly changing features in human reinforcement learning

Dear Dr Schuck,

I am pleased to inform you that your manuscript has been formally accepted for publication in PLOS Computational Biology. Your manuscript is now with our production department and you will be notified of the publication date in due course.

With kind regards,

Livia Horvath
